# Antarctic Ice Shelf Thickness Change from Multi-Mission Lidar Mapping

Tyler C. Sutterley[1], Thorsten Markus[1], Thomas A. Neumann[1], Michiel van den Broeke[2], J. Melchior van Wessem[2], and Stefan R. M. Ligtenberg[2]

[1]NASA Goddard Space Flight Center, Greenbelt, MD 20771
[2]Institute for Marine and Atmospheric Research, Utrecht University, Utrecht, The Netherlands

**Correspondence:** Tyler C. Sutterley (tyler.c.sutterley@nasa.gov)

**Abstract.**

We calculate rates of ice thickness change and bottom melt for ice shelves in West Antarctica and the Antarctic Peninsula from a combination of elevation measurements from NASA/CECS Antarctic ice mapping campaigns and NASA Operation IceBridge corrected for oceanic processes from measurements and models, surface velocity measurements from synthetic aperture radar, and high-resolution outputs from regional climate models. The ice thickness change rates are calculated in a Lagrangian reference frame to reduce the effects from advection of sharp vertical features, such as cracks and crevasses, that can saturate Eulerian-derived estimates. We use our method over different ice shelves in Antarctica, which vary in terms of size, repeat coverage from airborne altimetry and dominant processes governing their recent changes. We find that the Larsen-C Ice Shelf is close to steady state over our observation period with spatial variations in ice thickness largely due to the flux divergence of the shelf. Firn and surface processes are responsible for some short-term variability in ice thickness of the Larsen-C Ice Shelf over the time period. The Wilkins Ice Shelf is sensitive to short time-scale coastal and upper-ocean processes, and basal melt is the dominate contributor to the ice thickness change over the period. At the Pine Island Ice Shelf in the critical region near in the grounding zone, we find that ice shelf thickness change rates exceed 40 m/yr with the change dominated by strong submarine melting. Regions near the grounding zones of the Dotson and Crosson Ice Shelves are decreasing in thickness at rates greater than 40 m/yr, also due to intense basal melt. NASA/CECS Antarctic ice mapping and NASA Operation IceBridge campaigns provide validation datasets for floating ice shelves at moderately high resolution when co-registered using Lagrangian methods.

## 1 Introduction

Most of the drainage from the Antarctic ice sheet is through its peripheral ice shelves, floating extensions of the land ice that cover 75% of the Antarctic coastline and represent 10% of the total ice covered area (Cuffey and Paterson, 2010; Rignot et al., 2013). The modern-day thinning of Antarctic ice shelves may make the shelves more susceptible to fracture and overall collapse (Shepherd et al., 2003; Fricker and Padman, 2012). The mass budget of an ice shelf is the sum of several mass gain and loss terms (Thomas, 1979). Mass is gained by the advection of ice from the land, the accumulation of snow at the surface,

and the freezing of seawater at the ice shelf base (Thomas, 1979). Mass is lost by the runoff of surface meltwater, the erosion and sublimation of snow by wind, the sublimation of snow at the surface of the shelf, the melting of ice at the base of the shelf, and the calving of icebergs (Thomas, 1979).

Floating ice shelves can exert control on the grounded ice sheet's overall stability by buttressing the flow of the glaciers upstream (Dupont and Alley, 2005). The response of inland glaciers to ice shelf variations is complicated, and is dependent on both the inland bed topography and the ice shelf geometry (Goldberg et al., 2009; Gagliardini et al., 2010; Gudmundsson, 2013). Presently, several ice shelves across Antarctica are losing mass, which may have led to the acceleration and intensified discharge of inland ice (Pritchard et al., 2012; Depoorter et al., 2013; Paolo et al., 2016). In 2003, a year after the collapse of the Larsen-B Ice Shelf, some tributary glaciers draining into the Weddell Sea from the Antarctic Peninsula flowed at rates 2–8 times their 1996 flow rates (Rignot et al., 2004). These glaciers continued flowing at the accelerated rates several years after the collapse (Rignot et al., 2008; Berthier et al., 2012). Glaciers of the Amundsen Sea Embayment (ASE) in West Antarctica have experienced significant increases in surface velocity, dynamic thinning, and grounding line retreat since the 1990's (Rignot et al., 2002, 2014; Pritchard et al., 2009; Flament and Rémy, 2012). The dynamical change of these glaciers likely stems from increases in sub-shelf circulation and heat content of warm Circumpolar Deep Water, which enhanced ocean-driven melt causing thinning of the buttressing peripheral ice shelves (Jacobs et al., 2011).

Here, we compile ice shelf thickness change rates calculated using a suite of airborne altimetry datasets, which have been consistently processed and co-registered. We provide a set of co-registered laser altimetry datasets for evaluating estimates from satellite altimetry, photogrammetry and model outputs. The main objectives of this study are to (i) calculate ice shelf thickness change rates, (ii) investigate processes driving the changes in the shelf, (iii) investigate the sensitivity of spatial and temporal sampling to overall estimates and (iv) evaluate different methods of calculating elevation change rates over ice shelves. In the following sections, we discuss the co-registration method, the geophysical corrections applied, the results for a sample set of ice shelves and the overall implications of the results for ice shelf studies.

## 2   Materials and Methods

Our airborne lidar measurements are Level-2 Airborne Topographic Mapper (ATM) Icessn and Land, Vegetation and Ice Sensor (LVIS) datasets provided by the National Snow and Ice Data Center (NSIDC) (Thomas and Studinger, 2010; Studinger, 2014; Blair and Hofton, 2010). ATM is a conically scanning lidar developed at the NASA Wallops Flight Facility (Thomas and Studinger, 2010). ATM instruments have flown in Antarctica since 2002 as part of both NASA/Centro de Estudios Científicos (CECS) Antarctic ice mapping and NASA Operation IceBridge campaigns. The Level-2 ATM Icessn data is calculated by fitting planar surfaces to the original ATM point clouds at approximately 40 m spacing along track (Studinger, 2014). LVIS is a large-swath scanning lidar which flew in Antarctica in 2009, 2010, 2011 and 2015 and was developed at NASA Goddard Space Flight Center (Blair et al., 1999; Hofton et al., 2008). For the data release available for Antarctica (LDSv1), the Level-2 LVIS data provides 3 different elevation surfaces computed from the Level-1B waveforms: the highest and lowest returning surfaces from Gaussian decomposition, and the centroidal surface (Blair and Hofton, 2010). Here, we use the lowest returning

surface when the waveform resembles a single-peak gaussian and the centroid surface when the waveform is multi-peak. The spatial coverages of each instrument in Antarctica for the campaigns prior to and during NASA Operation IceBridge are shown in Figure 1. The elevation datasets from each instrument are converted to the 2014 solution of the International Terrestrial Reference Frame (ITRF) (Altamimi et al., 2016). In order to track changes in ice shelf freeboard, the ellipsoid heights for each instrument were converted to be in reference to the GGM05 geoid using gravity model coefficients provided by the Center for Space Research (Ries et al., 2016). Changes in ice shelf freeboard are converted into changes in ice thickness by assuming hydrostatic equilibrium following Fricker et al. (2001). Uncertainties for each instrument were calculated following Sutterley et al. (2018).

## 2.1 Integrated analysis of altimetry

We calculate rates of elevation change by comparing a set of measured elevation values with a set of interpolated elevation values from a different time period after allowing for the advection of the ice (Sutterley et al., 2018; Moholdt et al., 2014; Shean et al., 2018). Each point in a flight line is advected from its original location by integrating the Rignot et al. (2017) MEaSUREs static velocity data derived from synthetic aperture radar (SAR) using a fourth-order Runge-Kutta algorithm. For each data point in a flight line, a set of Delaunay triangles is constructed from a separate flight line using all data points within 300 m from the final location of the advected point (Pritchard et al., 2009, 2012; Rignot et al., 2013). If the advected point lies within the confines of the Delaunay triangulation convex hull, the triangular facet housing the advected point is determined using a winding number algorithm (Sutterley et al., 2018). The new elevation value is calculated using barycentric interpolation with the elevation measurements at the three triangle vertices (Figure 2). The elevation at each vertex point is weighted in the interpolation by the area of the triangle created by the enclosed point and the two opposing vertices (Sutterley et al., 2018).

Assuming that the ice shelf surfaces are not curved over the scale of the individual triangular facet ($\sim$10–100 m), interpolating to the advected coordinates will compensate for minor slopes in the ice shelf surface so that the elevations of equivalent parcels of ice can be compared in time (Pritchard et al., 2009). At this scale (below 100–200 m), the topographic relief of uncrevassed ice is primarily due to slopes in the ice surface and a planar assumption should be largely valid (Markus et al., 2017). Rough terrain, snow drifts and low-lying clouds will contaminate the lidar elevation values for the interpolation. In order to limit the effect of contaminated points, the elevation measurements are filtered using the Robust Dispersion Estimator (RDE) algorithm described in Smith et al. (2017). In order to minimize the possibility of co-registering measurements over ice shelves with measurements over grounded ice near the grounding zone or measurements over the ocean, sea ice floes and icebergs, we only include points that are on the ice shelf for the compared time periods using grounded ice delineations from Rignot et al. (2016) and Mouginot et al. (2017b) and ice shelf extent delineations manually digitized from Landsat imagery courtesy of the U.S. Geological Survey and MODIS imagery from Scambos et al. (2001).

For comparison, we compile elevation change measurements using an Eulerian approach with the Triangulated Irregular Networks (TINs) technique outlined in Sutterley et al. (2018) and a Lagrangian overlapping footprint approach following Slobbe et al. (2008) and Moholdt et al. (2014). The Eulerian TINs scheme follows the methods of Pritchard et al. (2012) and Rignot et al. (2013) that used data from the NASA ICESat mission. Measurements compiled using the Eulerian TINs scheme

have been made comparable to the Lagrangian thinning rates by adding the effects of strain using the relation from Moholdt et al. (2014).

$$\frac{Dh}{Dt} = \frac{\partial h}{\partial t} + \frac{\rho_w - \rho_{ice}}{\rho_w \rho_{ice}} V \cdot \nabla M \tag{1}$$

where $\rho_w$ and $\rho_{ice}$ are the densities of sea water and meteoric ice, respectively, and $(V \cdot \nabla M)$ is the ice shelf thickness gradient advection. For calculating the mass divergence for comparing Eulerian and Lagrangian-derived ice thickness change rates, we use ice thickness data and uncertainties from Bedmap2, which are primarily derived from Griggs and Bamber (2011) for ice shelves (Fretwell et al., 2013). The ice thickness data from Griggs and Bamber (2011) are calculated assuming hydrostatic equilibrium, which should be valid for most areas downstream of the 1–8 km wide grounding zones (Brunt et al., 2010, 2011). The Lagrangian overlapping footprint approach uses the same fourth-order Runge-Kutta algorithm to advect the coordinates of the original elevation measurement to a predicted parcel location at a separate time. If any measurements from the separate flight line lie within 100 m of the advected point, the elevation measurement closest in Euclidean distance to the advected point is compared against the original measurement.

## 2.2 Geophysical Corrections

We correct the elevation measurements for geophysical processes following most of the procedures that will be used with the initial release of ICESat-2 data (Markus et al., 2017; Neumann et al., 2018). The processes are described in the following sections and represented as a schematic in Figure 3.

### 2.2.1 Tidal and Non-Tidal Ocean Variation

Surface elevation changes due to variations in ocean and load tides are calculated using outputs from the Circum-Antarctic Tidal Simulation (CATS2008) model (Padman et al., 2008), a high-resolution inverse model updated from Padman et al. (2002). Surface heights were predicted for the $M_2$, $S_2$, $N_2$, $K_2$, $K_1$, $O_1$, $P_1$, $Q_1$, $M_f$ and $M_m$ harmonic constituents and then inferred for 16 minor constituents following the *PERTH3* algorithm developed by Richard Ray at NASA Goddard Space Flight Center (Ray, 1999). Uncertainties in tidal oscillations were estimated using constituent uncertainties from King et al. (2011). We correct for changes in load and ocean pole tides due to changes in the Earth's rotation vector following Desai (2002) and IERS conventions (Petit and Luzum, 2010). We correct for changes in sea surface height due to changes in atmospheric pressure and wind stress using a dynamic atmosphere correction (DAC) provided by AVISO. The 6-hour DAC product combines outputs of the MOD2D-g ocean model, a 2-D ocean model forced by pressure and wind fields from ECMWF based on Lynch and Gray (1979), with an inverse barometer (IB) response (Carrère and Lyard, 2003). Regional sea levels fluctuate due to changes in ocean dynamics, ocean mass, and ocean heat content (Church et al., 2011; Armitage et al., 2018). Non-tidal sea surface anomalies are removed from the ice shelf data using multi-mission altimetry products computed by AVISO and provided by Copernicus (Le Traon et al., 1998). The non-tidal sea surface anomalies are added to estimates of mean dynamic topography, which is the mean deviation of the sea surface from the Earth's geoid due to ocean circulation. The sea surface anomalies are extrapolated from the valid ice-free ocean values to the ice shelf points following Paolo et al. (2016).

### 2.2.2 Surface Mass Balance and Firn Compaction

After correcting for the effects of oceanic variation and advection, changes in surface height are due to a combination of accumulation, ablation and firn densification processes. To account for variations in surface elevation due to changes in surface processes, we use monthly mean surface mass balance (SMB) outputs calculated from climate simulations of the Regional

Atmospheric Climate Model (RACMO2.3p2) computed by the Ice and Climate group at the Institute for Marine and Atmospheric Research of Utrecht University (Ligtenberg et al., 2013; van Wessem et al., 2014, 2018). We use 5.5km horizontal resolution outputs from a 1979–2016 climate simulation of the Antarctic Peninsula (XPEN055, van Wessem et al., 2016) and a 1979–2015 climate simulation of West Antarctica (ASE055, Lenaerts et al., 2018). The high-resolution outputs better represent the surface mass balance state than outputs from the 27km ice sheet wide model, particularly in the highly complex topography

of mountains and glacial valleys in the Antarctic peninsula (van Wessem et al., 2016). The higher spatial resolution topography improves the modeling of wind-driven downstream effects over ice shelves (Datta et al., 2018). SMB is the quantified difference between mass inputs from the precipitation of snow and rain, and mass losses by sublimation, runoff, and wind scour (Lenaerts et al., 2012; van den Broeke et al., 2009). Runoff is the portion of total snowmelt not retained or refrozen within the ice sheet. Wind scour is the erosion and sublimation of wind-blown snow from the ice sheet surface (Das et al., 2013).

The absolute precision of the RACMO2.3p2 model outputs has been estimated using NASA Operation IceBridge snow radar observations, satellite observations of surface melt, and and in-situ observations, such as ice cores and surface stake measurements, following Kuipers Munneke et al. (2017) and Lenaerts et al. (2018). To correct for variations in the firn layer thickness, we use air content outputs from a semi-empirical firn densification model that simulates the steady-state firn density profile (Ligtenberg et al., 2011, 2012). The firn densification model is forced with surface mass balance outputs, surface temperatures fields and near-surface wind speed fields computed by RACMO2.3p2 (Ligtenberg et al., 2011). We assume a 15% uncertainty

in surface mass balance and firn air content height change following estimates from Kuipers Munneke et al. (2017).

### 2.3 Ice Shelf Bottom Melt

Changes in ice shelf mass in a Lagrangian reference frame are due to changes in surface mass balance (SMB) processes ($M_s$), basal melt ($M_b$) and the divergence of the ice shelf flow field ($M\nabla \cdot V$) (Moholdt et al., 2014).

$$\frac{dM_s}{dt} + \frac{dM_b}{dt} - M\nabla \cdot V = \frac{\rho_w \rho_{ice}}{\rho_w - \rho_{ice}} \left( \frac{Dh}{Dt} - \frac{\partial h_{oc}}{\partial t} - \frac{\partial h_{fc}}{\partial t} \right) \tag{2}$$

where $h_{oc}$ are ocean heights, and $h_{fc}$ are firn-column air content heights. We estimate ice shelf bottom melt rates along flight lines by using mass conservation and estimates of the mass flux divergence (Rignot and Jacobs, 2002; Moholdt et al., 2014; Rignot et al., 2013). Ice flow divergence fields are calculated from ice velocities from Rignot et al. (2017) differentiated using a Savitzky-Golay filter with an 11 km half-width window (Savitzky and Golay, 1964). The Savitzky-Golay algorithm

smooths the velocity field, and reduces the impact of ionospheric noise and other sources of uncertainty on the differentials. Deviations from mean ice flow were calculated using annually resolved ice velocity maps derived from synthetic aperture radar and optical imagery (Mouginot et al., 2017a). Ice shelf masses were calculated by converting the altimetry-derived ice shelf freeboard heights to ice thickness by assuming hydrostatic equilibrium (Fricker et al., 2001; Griggs and Bamber, 2011).

## 3 Results

We co-register 134 days of ATM data and 32 days of LVIS data for the years 2002–2016. We compare elevation change measurements between Eulerian and Lagrangian approaches derived using Triangulated Irregular Networks (TINs) (Sutterley et al., 2018, Figure 4). Using a Lagrangian reference frame can produce estimates of ice shelf elevation change with much less
noise compared with a Eulerian reference frame (Moholdt et al., 2014, Figure 4). This is because the advection of ice thickness gradients, such as that from cracks and crevasses in the ice, can saturate the Eulerian-derived estimates (Moholdt et al., 2014; Shean et al., 2018).

### 3.1 Larsen Ice Shelves

The ice shelves draining from the Antarctic Peninsula into the Weddell Sea have undergone some significant changes over the
past three decades. The Larsen-A Ice Shelf collapsed in 1995, and the Larsen-B Ice Shelf partially collapsed in 2002 (Rott et al., 2002, 2011). The tributary glaciers once flowing into these shelves accelerated with the loss of the ice shelf abutment (Rignot et al., 2008). We estimate the impact of surface processes, ice divergence, and basal melt using data from a flight line starting near the Whirlwind Inlet of the Larsen-C Ice Shelf (Figure 5a). Scatter in the Lagrangian-derived ice thickness change, $DH/Dt$, across the flight line is typically 30–50 cm/yr, or a 4–6 cm/yr error in the measured elevation change rate (Figure 5a).
Most of the thickness change, $DH/Dt$, along this line is due to the flux divergence of the shelf, indicating the shelf along this line is nearly in steady-state during this period. As the basal melt rate is calculated via mass conservation and the estimated $DH/Dt$ rate largely matches the flux divergence, estimates of the basal melt rate of the Larsen-C Ice Shelf are highly dependent on the SMB flux estimate. Any uncertainties in reconstructing the regional SMB will significantly impact the resultant basal melt rate estimate. The $DH/Dt$ rate of the Larsen-B Remnant and Larsen-C Ice Shelves for two periods, 2002–2008 and
2008–2016, from NASA/CECS Pre-IceBridge and NASA Operation IceBridge airborne data is shown in Figure 6 (a-b). The estimated basal melt rate of the ice shelves over the same periods is shown in Figure 6 (c-d). The average $DH/Dt$ rate between 2008 and 2016 from the flight line data over the Larsen-C Ice Shelf is –1.2±0.9 m/yr. From 2008–2016, the strongest $DH/Dt$ rates occur near the grounding zone, particularly for the flight lines starting near Cabinet and Mill Inlets. We compare our airborne laser altimetry estimate of basal melt rates with a long-term record derived from radar altimetry (Adusumilli et al.,
2018). We find that the radar-derived estimate is comparable with the laser-derived estimate within uncertainties for most points outside of the grounding zone (Figure 7). However, due to the sensitivity of the laser altimetry estimate to the SMB model (Figure 5a), measurements from radar altimetry may be more accurate determinations of basal melt rate for the ice shelf.

### 3.2 Wilkins Ice Shelf

The Wilkins Ice Shelf is fed by glaciers on Alexander Island, which is located near the west coast of the Antarctic Peninsula and is the largest of the Antarctic islands. Wilkins Ice Shelf is sensitive to short time-scale coastal and upper-ocean processes (Padman et al., 2012) and ablates largely through basal melting (Rignot et al., 2013). $DH/Dt$ (a-b) and estimated basal

melt rates (c-d) of the Wilkins Ice Shelf for two 3-year periods from 2008–2011 and 2011–2014 is shown in Figure 8. The extent of the ice shelf reduced from 16000 to 10000 km$^2$ (38%) between 1990 and 2017 (Scambos et al., 2009). The partial collapse occurred once the shelf started decoupling from Charcot Island (Vaughan et al., 1993) and likely occurred due to hydro-fracturing (Scambos et al., 2009). Meltwater ponds covered areas of 300–600 km$^2$ in Landsat imagery in 1986 and 1990

(Vaughan et al., 1993). The ponds existed largely in the now-collapsed portions of the shelf near Rothschild Island. Average $DH/Dt$ rates of the Wilkins Ice Shelf from the flight lines were –1.3±0.7 m/yr from 2008–2011 and –0.7±0.5 m/yr from 2011–2014. Average estimated basal melt rates from the flight lines were 2.5±1.3 m/yr in the earlier period and 1.8±0.9 m/yr in the latter period. Basal accretion could have occurred in some regions during the 2011–2014 period.

## 3.3 Pine Island Ice Shelf

The Pine Island Ice Shelf abuts one of the most rapidly changing glaciers in Antarctica (Pritchard et al., 2009; Flament and Rémy, 2012). Figure 9 shows the change in ice thickness (a-b) and estimated basal melt rates (c-d) of the Pine Island Ice Shelf for two periods from 2009–2011 and 2011–2015. These periods were chosen to include repeat measurements from LVIS of the ice shelf near the grounding zone and to use the high-resolution outputs of RACMO2.3p2 ASE055. The average $DH/Dt$ rates from the flight lines were insignificantly different at –35±9 m/yr over 2009–2011 and –33±5 m/yr over 2011–2015.

Because basal melt rates near the grounding zone have the highest impact on the glacial flow dynamics, we estimate the basal melt rate between the 1996 and 2011 grounding lines (Rignot and Jacobs, 2002). In this previously grounded region, the ice shelf $DH/Dt$ rates were –96±15 m/yr during 2009–2011 and –79±7 m/yr during 2011–2015. In this area that was previously grounded, the average estimated basal melt rates from the flight lines were 77±18 m/yr over 2009–2011 and 61±12 m/yr over 2011–2015. $DH/Dt$ rates outside of the previously grounded area between the 1996 and 2011 grounding lines

are significantly weaker than in the previously grounded area, averaging –20±7 m/yr for 2009–2011 and –15±3 m/yr for 2011–2015. The difference in melt rates near the grounding zone between 2009–2011 and 2011–2015 could possibly explain some of the moderation in thinning of the grounded ice and stability in ice discharge from Pine Island Glacier after 2010 (McMillan et al., 2014; Medley et al., 2014). As shown in Figure 9c-d, the $DH/Dt$ rate is dominated by strong submarine melt, which is further evidence of the dominant oceanic controls on the ice shelf mass balance in this region (Rignot, 2002;

Shean et al., 2018). However, some of the changes in basal melt rate over the period could be due to large regional interannual-to-decadal variability (Dutrieux et al., 2014; Paolo et al., 2015; Jenkins et al., 2018). We compare our estimates of Pine Island Ice Shelf change from airborne laser altimetry with ICESat-derived surface elevation change from Pritchard et al. (2012) and basal melt rate from Rignot et al. (2013) (Figures 10 and 13a-b). While there are few data points for comparison and the time periods are not contemporaneous (2002–2009 for the airborne data and 2003–2009 for the ICESat data), we find some

significant differences between our airborne altimetry-derived estimates and the satellite derived estimates (Figure 10c,f). The RMS difference between the airborne-derived estimate and the satellite-derived estimates are 31 m/yr in terms of basal melt rate (Rignot et al., 2013) and 8 m/yr in terms of surface elevation change (Pritchard et al., 2012). For the coincident data, the airborne altimetry data showed more variability in basal melt rate and surface elevation change than the satellite-derived methods (Figure 13a-b). The differences in variability are likely due to the different spatial resolutions of the datasets, the

different geophysical corrections applied for each estimate, and the spatial smoothing applied to the Pritchard et al. (2012) and Rignot et al. (2013) estimates.

## 3.4 Dotson and Crosson Ice Shelves

The glaciers flowing into the Dotson and Crosson Ice Shelves have rapidly thinned, increased in speed and experienced significant retreats of grounding line positions over the past 20 years (Mouginot et al., 2014; Scheuchl et al., 2016). Flow speeds of the Crosson Ice Shelf have doubled in some regions over 1996 to 2014, while the flow speed of Dotson has remained largely steady (Lilien et al., 2018). $DH/Dt$ rates (a-b) and estimated basal melt rates (c-d) of the Dotson and Crosson Ice Shelves are shown in Figure 11 for two periods, 2002–2010 and 2010–2015. Regions near the grounding lines of the input glaciers are decreasing in thickness rapidly for both shelves driven by strong basal melt. Basal melt rates averaged 47–81 m/yr near the grounding zone of Smith glacier over the two periods. Khazendar et al. (2016) documented rapid submarine ice melt and the loss of 300–490 m of floating ice between 2002 and 2009. Our work here provides independent evidence of this large-scale melt using a separate method and more years of data. We find that the ice mass wastage continued unabated between 2010 and 2015 with $DH/Dt$ rates over the flight lines averaging –21±1 m/yr. We compare our airborne laser altimetry data of the Dotson and Crosson Ice Shelves with satellite laser altimetry estimates of surface elevation change from Pritchard et al. (2012) and basal melt rate from Rignot et al. (2013) (Figures 12 and 13c-d). The RMS difference between the airborne-derived estimate and the satellite-derived estimates are 5 m/yr in terms of basal melt rate (Rignot et al., 2013) and 4 m/yr in terms of surface elevation change (Pritchard et al., 2012). For the coincident data, the airborne altimetry data aligns well with the satellite-derived estimate of basal melt rate from Rignot et al. (2013) (Figure 13c). However, the surface elevation estimates from Pritchard et al. (2012) do not align well with our the airborne altimetry-derived estimate (Figure 13d). The difference is likely due to the lack of spatial coverage with the airborne estimate, which may not be representative at the 10 km horizontal spatial scale of the Pritchard et al. (2012) estimate, particularly closer to the grounding line (Figure 12f).

## 4  Discussion

Using a Lagrangian reference frame may result in fewer co-registered data points and less spatial coverage of measurements compared with using an Eulerian reference frame (Figure 4). Lagrangian tracking of airborne data requires 1) accurate flow-line flight planning, 2) a sufficiently wide scanning swath, or 3) dense grid measurements. Flight lines along-flow need to be accurately planned to ensure upstream measurements can be paired with future downstream measurements. With the current airborne data at most locations, cross-flow flight lines advected outside of the swath width over multi-year repeat times. This limited our dataset to regions with flow-line measurements, such as the Larsen-C Ice Shelf (Figure 6), or frequent measurements, such as the Dotson and Crosson Ice Shelves (Figure 11). For most ice shelves, repeated airborne data is too sparse to extract large-scale spatial trends, particularly in the era before NASA Operation IceBridge. Satellite altimetry measurements from the NASA ICESat-2 mission (Markus et al., 2017) should help rectify the data limitation problem by providing dense and repeated point clouds. ICESat-2 data could be combined with photogrammetric digital elevation models (DEMs) to cre-

ate high-resolution ice shelf-wide thickness change maps (Berger et al., 2017; Shean et al., 2018). Combining ICESat-2 with DEMs would help improve the use of the laser altimetry data in a Lagrangian reference frame as ice parcels could be accurately tracked between separate satellite tracks. In addition, isolated crossovers can be calculated with the airborne data using Lagrangian tracking for some ice shelves using along-flow and cross-flow measurements from separate years. These singular

crossovers would likely not be representative of the large-scale behavior of the ice shelf due to the spatial variability of ice thickness change, but may still provide valuable metrics for evaluating outputs from ice sheet models (Figures 10 and 12).

Lagrangian-derived estimates also greatly depend on the quality of the velocity estimates used for advecting the ice parcels in time. Here, the airborne data are co-registered in a Lagrangian reference frame using a static velocity map provided by NSIDC through the MEaSUREs program (Rignot et al., 2017). However, there are cases that do not fit the assumption of

temporally-invariant velocities. Prior to the calving event of the 40,000 km$^2$ A-68 iceberg from the Larsen-C Ice Shelf on July 11, 2017, the ice was rifting from the south and the regions downstream of the crack were rotating outward (Hogg and Gudmundsson, 2017, Figure 6). In the Amundsen Sea Embayment, the ice velocity structure has changed year-to-year since the 1990's (Rignot et al., 2008; Mouginot et al., 2014). The floating ice shelves in the Amundsen Sea are also rifting concurrently with the acceleration of the instreaming glaciers (Macgregor et al., 2012). For both of these cases, it would be more appropriate

to predict the advected parcel location using time-variable velocity maps. However, the spatial coverage of annual velocity maps is lacking for some time periods, which will complicate the advection calculation. For some locations, such as near shear margins, ice velocities can vary at smaller spatial scales than what is presently available from SAR measurements and visible imagery feature-tracking. With the high-temporal resolution data from the ESA Sentinel mission, the Landsat-based goLIVE project and the future NASA-ISRO SAR mission (NISAR), the advected parcel locations could be predicted with

much greater accuracy for recent NASA Operation IceBridge campaigns and future altimetry missions (Fahnestock et al., 2016; Gardner et al., 2018; Mouginot et al., 2017a). Improvements in ice thickness and ice velocity estimates will also greatly improve estimates of flux divergence and as a consequence estimates of basal melt rates calculated using mass conservation (Berger et al., 2017; Adusumilli et al., 2018).

This work builds off of the work of Paolo et al. (2015) and Adusumilli et al. (2018) that used radar altimetry data to analyze

the recent thinning and basal melt rates of ice shelves. Paolo et al. (2015) calculated changes in the ice thickness time series over an 18-year time period using a suite of satellite radar altimetry data compiled in an Eulerian frame of reference. They found that the overall volume loss of ice shelves accelerated over the period 1994–2012, particularly for the ice shelves of West Antarctica. Adusumilli et al. (2018) expanded on this work by including radar altimetry data from CryoSat-2 to estimate the basal melt rates in the Antarctic Peninsula over a 23 year period. Laser altimeters and radar altimeters can measure different

surfaces over snow-covered ice surfaces (Rémy and Parouty, 2009). Idealistically, the laser altimeter will detect the snow surface and the radar altimeter will detect the snow-ice interface. Because laser altimeters ideally detect the snow surface, an estimate of the total column snow/firn height change is needed to calculate the ice shelf freeboard change (Pritchard et al., 2012). For radar altimeters, the radar penetration depth is affected by variations in the dielectric properties of the surface layer due to variations in temperature, snow grain size, snow density and moisture content (Partington et al., 1989; Rémy and

Parouty, 2009). Due to the variations in penetration depth, estimates of the firn height change below the detected surface are

necessary in order to calculate the freeboard change. Determining the sensitivity of radar estimates to surface penetration over different surface types could help reconcile differences between the various estimates (Figure 7). In addition, in regions of uncertain surface mass balance and firn change, inter-comparisons with radar altimetry estimates may help provide important metrics for improving SMB and firn models. In these regions, radar altimetry estimates of ice thickness change may be more

accurate than from laser altimetry due to the SMB uncertainty.

Compiling estimates of elevation change from laser altimetry is non-trivial and different processing methods can produce differing results. Felikson et al. (2017) compared four different processing schemes (crossover differencing, along-track surface fits, overlapping footprints and triangulated irregular networks) using ICESat data in an Eulerian sense over grounded ice in Greenland. They found discernible and irreconcilable differences between methods when deriving elevation change over

the grounded ice sheet. We compare results from overlapping footprints and triangulated irregular networks to test their correspondence over ice shelf surfaces. As the surface slopes on ice shelves are small, we find that overlapping footprints and TINs approaches produce similar estimates of elevation change with scanning lidars in Lagrangian frames of reference (Figure 4). The overlapping footprints approach produces a slightly noisier but statistically similar estimate compared with the TINs approach, and is a significantly simpler algorithm to implement.

**5   Conclusions**

We present a method for measuring ice shelf thickness change through the co-registration of NASA/CECS Antarctic ice mapping and NASA Operation IceBridge laser altimetry data in a Lagrangian reference frame. We use our method to detect changes in ice shelves in West Antarctica and the Antarctic Peninsula where the airborne data are available. We find that our method can be a significant improvement over Eulerian-derived estimates that may require substantial spatial averaging of the

data to reduce the impact of noise. However, there are significant limitations when using airborne data for detecting ice shelf thickness change with Lagrangian tracking, particularly the lower spatial coverage and typical lack of repeat tracks over ice shelves. Data from the recently launched NASA ICESat-2 mission will help rectify these problems, particularly if combined with high-resolution photogrammetric digital elevation models.

*Code and data availability.* NASA Operation IceBridge data are freely available from the National Snow and Ice Data Center (NSIDC) at

http://nsidc.org/data/ILATM2/ for the Level-2 ATM data and http://nsidc.org/data/ILVIS2/ for the Level-2 LVIS data. NASA MEaSUREs INSAR-derived velocity maps are provided by NSIDC at https://nsidc.org/data/nsidc-0484. Bedmap2 ice thicknesses are provided by the British Antarctic Survey at https://www.bas.ac.uk/project/bedmap-2/. CATS2008 tidal constituents are available from the Earth & Space Research institute at https://www.esr.org/research/polar-tide-models/. Dynamic atmospheric Corrections are produced by CLS Space Oceanography Division using the Mog2D model from Legos distributed by Aviso, with support from CNES. Ssalto/Duacs non-tidal sea surface

products were produced and distributed by the Copernicus Marine and Environment Monitoring Service (CMEMS). Landsat imagery is provided courtesy of the U.S. Geological Survey EarthExplorer service. MODIS images of ice shelves are freely available from NSIDC. Altimetry data from this project are available on Figshare under a CC BY 4.0 license (doi:10.6084/m9.figshare.8159852). The following pro-

grams are provided by this project for processing the NASA Operation IceBridge data: *nsidc-earthdata* retrieves NASA data from NSIDC (doi:10.6084/m9.figshare.7355063), *read-ATM1b-QFIT-binary* reads Level-1b Airborne Topographic Mapper (ATM) QFIT binary data products (doi:10.6084/m9.figshare.7355060), *read-ATM2-icessn* reads Level-2 ATM Icessn data products (doi:10.6084/m9.figshare.7355066), and *read-LVIS2-elevation* reads Level-2 Land Vegetation and Ice Sensor (LVIS) data products (doi:10.6084/m9.figshare.7355057).

*Author contributions.*  T.C.S. performed the analysis and wrote the manuscript. T.M. and T.A.N. supervised the project and provided comments and feedback. M.v.d.B, S.R.M.L. and J.M.v.W. supplied the RACMO2.3p2 data and provided comments.

*Competing interests.*  The authors declare that the research was conducted in the absence of any commercial or financial relationships that could be construed as a potential conflict of interest.

*Acknowledgements.*  Research was supported by an appointment to the NASA Postdoctoral Program at NASA Goddard Space Flight Center, administered by Universities Space Research Association under contract with NASA. The authors wish to thank our editor Kenichi Matsuoka, reviewer Laurence Padman, and our other anonymous reviewer for their comments and suggestions on improving this manuscript. We wish to thank Eric Rignot (UCI/JPL) and Isabella Velicogna (UCI/JPL) for their comments, and members of the GSFC Cryospheric Sciences Laboratory for their comments and advice. The authors thank Susheel Adusumilli (UCSD/SIO) for providing the estimated basal melt rates of the Larsen-C ice shelf from radar altimetry data, Hamish Pritchard (BAS) for providing the elevation change rates from ICESat altimetry data, and Jeremie Mouginot (UCI) for providing the estimated basal melt rates from ICESat altimetry data and his advice on the laser altimetry analysis. The authors wish to acknowledge the NASA Operation IceBridge flight, instrument and science teams for their work to collect and produce the science data. We would also wish to thank the National Snow and Ice Data Center (NSIDC) for storing and distributing the data from the NASA/CECS Antarctic ice mapping campaigns and NASA Operation IceBridge.

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

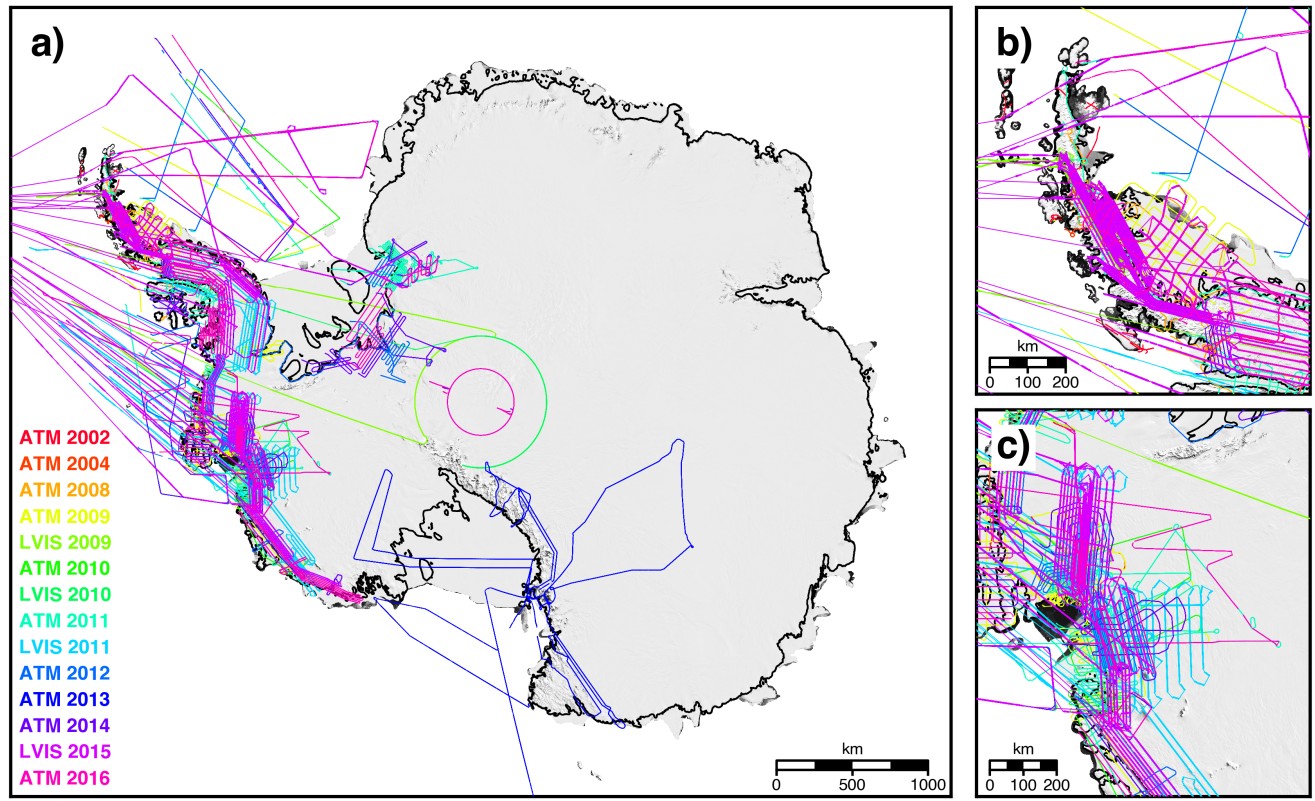

**Figure 1.** NASA/CECS Pre-IceBridge and NASA Operation IceBridge campaign flight lines over a) Antarctica b) the Antarctic Peninsula and c) the Amundsen Sea Embayment from 2002 to 2016 colored by year of acquisition and laser ranging instrument. Antarctic grounded ice delineation provided by Mouginot et al. (2017b). Flight lines overlaid on a 2008–2009 MODIS mosaic of Antarctica (Haran et al., 2014).

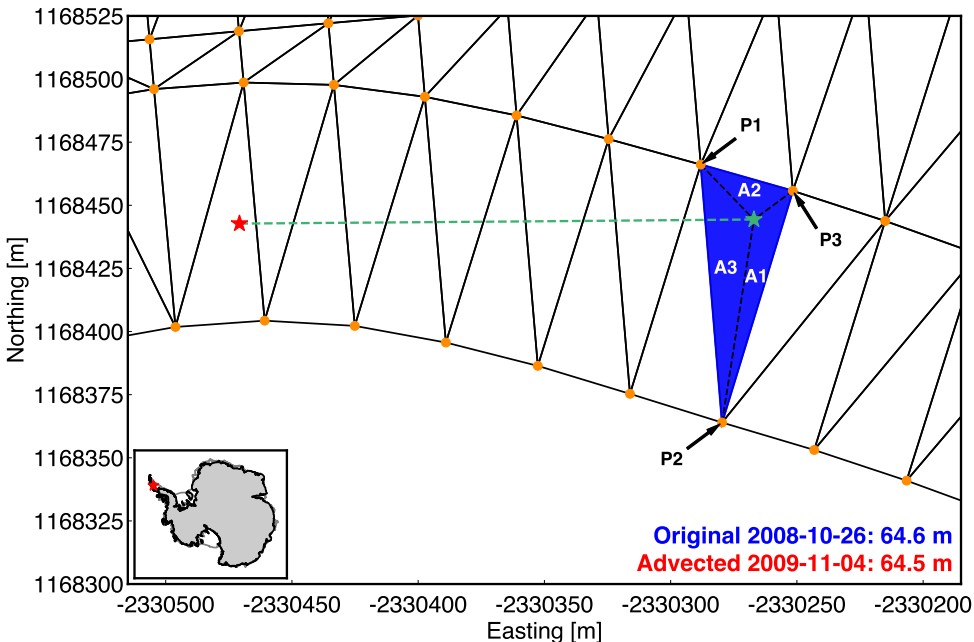

**Figure 2.** Triangulated mesh formulated around an advected 2008 ATM flight line point using points from a 2009 ATM flight line (orange dots). The red star denotes the location of the original point, the green star denotes the parcel location after advection, and the dashed green line is the path of advection. P1, P2 and P3 represent the three vertices of the triangle housing the advected ATM point. Elevation values at each vertex point are weighted in the interpolation by their respective areas, A1, A2 and A3. Inset map shows the location of the main figure.

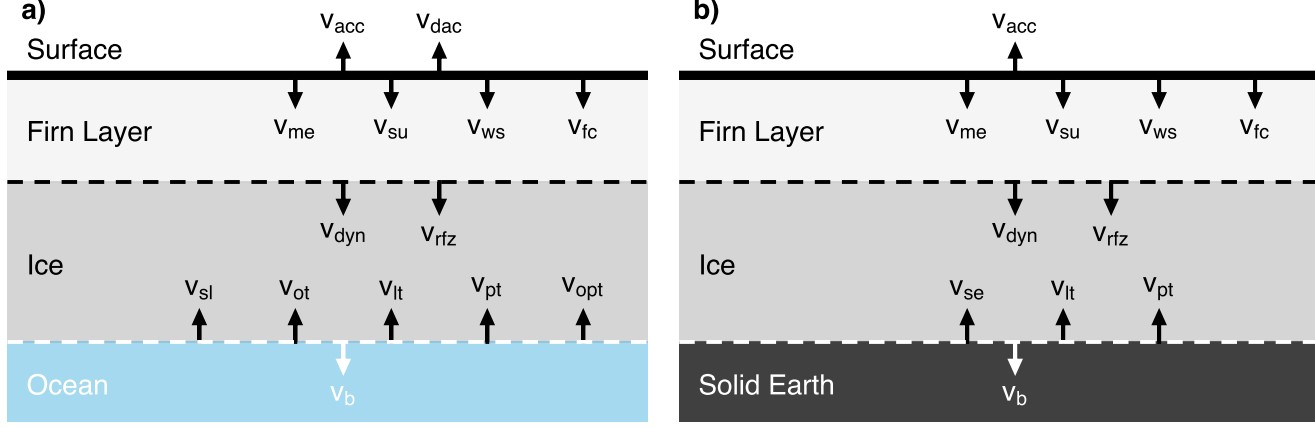

**Figure 3.** Representation of processes contributing to surface elevation changes for a) ice shelves and b) grounded ice. Modified from Ligtenberg et al. (2011) and Zwally and Li (2002). Processes represented in schematic: accumulation ($v_{acc}$), dynamic atmosphere ($v_{dac}$), snowmelt ($v_{me}$), sublimation ($v_{su}$), wind scour ($v_{ws}$), firn compaction ($v_{fc}$), ice dynamics ($v_{dyn}$), meltwater refreeze and retainment ($v_{rfz}$), solid Earth uplift ($v_{se}$), sea level ($v_{sl}$), ocean tides ($v_{ot}$), load tides ($v_{lt}$), load pole tides ($v_{pt}$), ocean pole tides ($v_{opt}$), and basal melt ($v_b$).

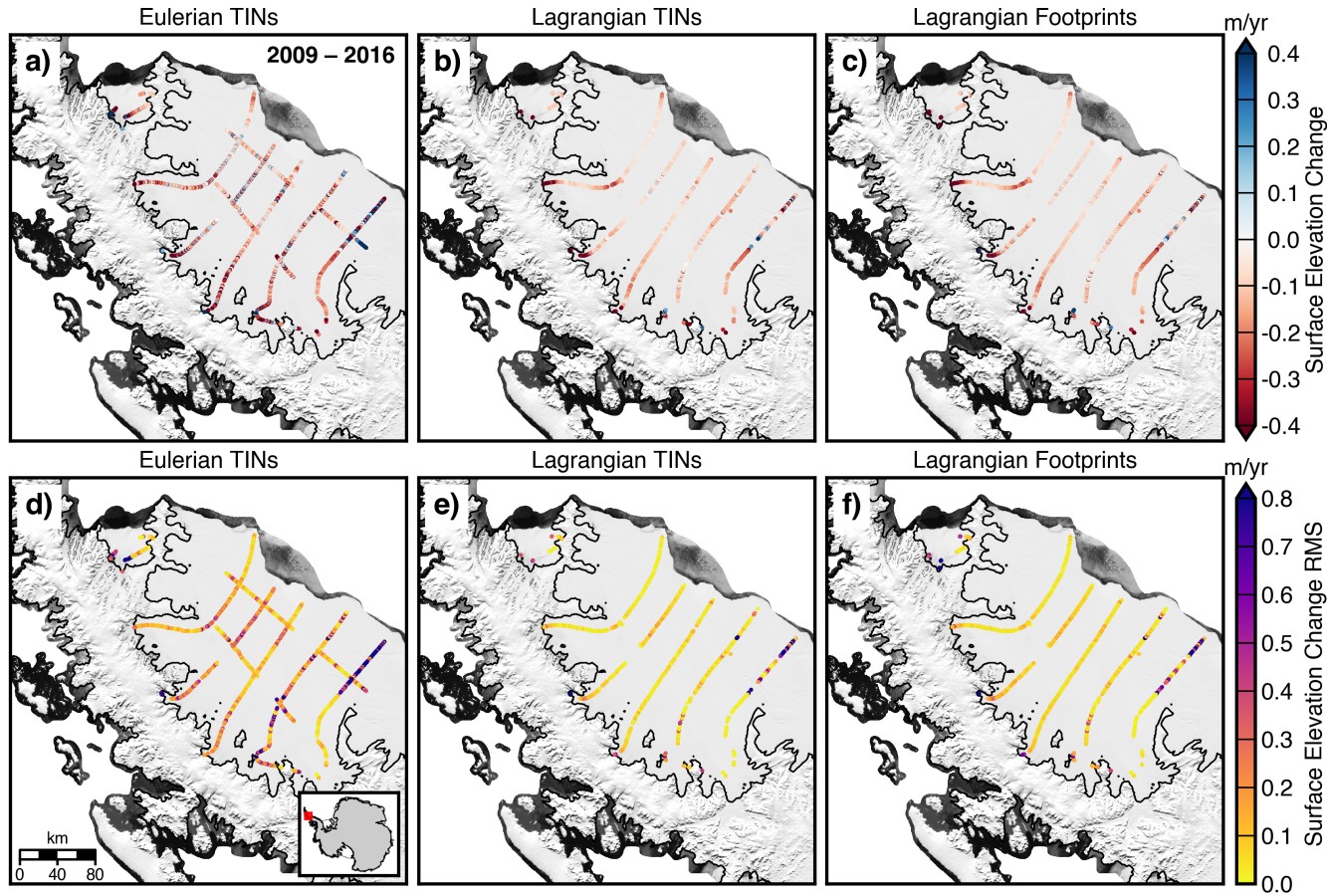

**Figure 4.** Surface elevation change of the Larsen-B remnant and Larsen-C Ice Shelf derived using a) Eulerian TINs corrected for strain, b) Lagrangian TINs and c) Lagrangian overlapping footprint schemes for the period 2009–2016. RMS differences in elevation change from a measurement point for all points within 1 km for the d) Eulerian TINs corrected for strain, e) Lagrangian TINs and f) Lagrangian overlapping footprint methods. The elevation change rates shown here are not RDE filtered (Smith et al., 2017). Antarctic grounded ice boundaries are provided by Mouginot et al. (2017b). Plots are overlaid on a 2008–2009 MODIS mosaic of Antarctica (Haran et al., 2014). Inset map denotes the location of the maps.

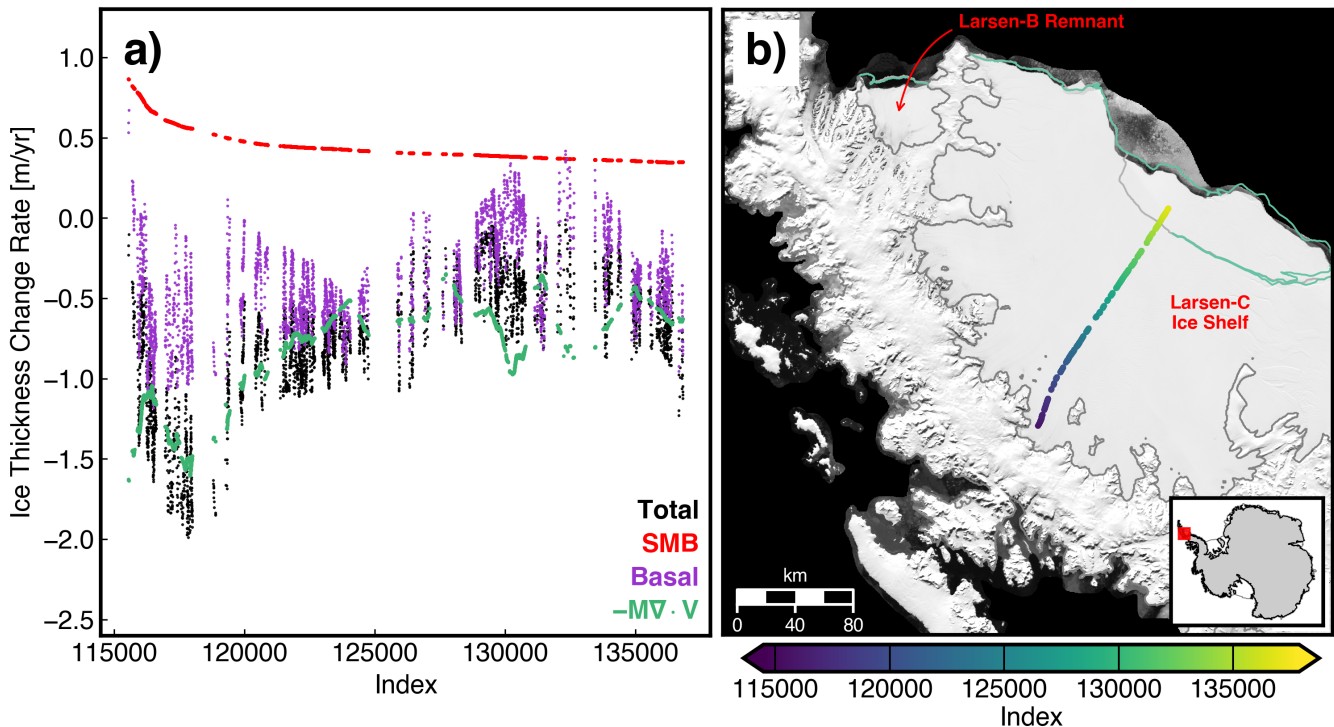

**Figure 5.** Measured and estimated ice thickness change rates from 2008 to 2016 for a flight line over the Larsen-C Ice Shelf (a) starting near the Whirlwind Inlet with the total measured ice thickness change rate denoted in black, the surface mass balance (SMB) fluxes from RACMO2.3p2 (XPEN055) denoted in red (van Wessem et al., 2016), the flux divergence terms combining ice velocities from MEaSUREs (Rignot et al., 2017)and ice thicknesses denoted in green, and the residual basal thickness change rates denoted in purple. Index denotes the ATM Icessn record number for October 10, 2008. Locations of co-registered records from the flight line are shown in b). MEaSUREs InSAR-derived Antarctic grounded ice boundaries are denoted in gray (Mouginot et al., 2017b). 2016 and 2017 ice shelf extents delineated from MODIS imagery are denoted in green and light gray, respectively (Scambos et al., 2001). Map is overlaid on a 2008–2009 MODIS mosaic of Antarctica (Haran et al., 2014). Inset map denotes the location of the map.

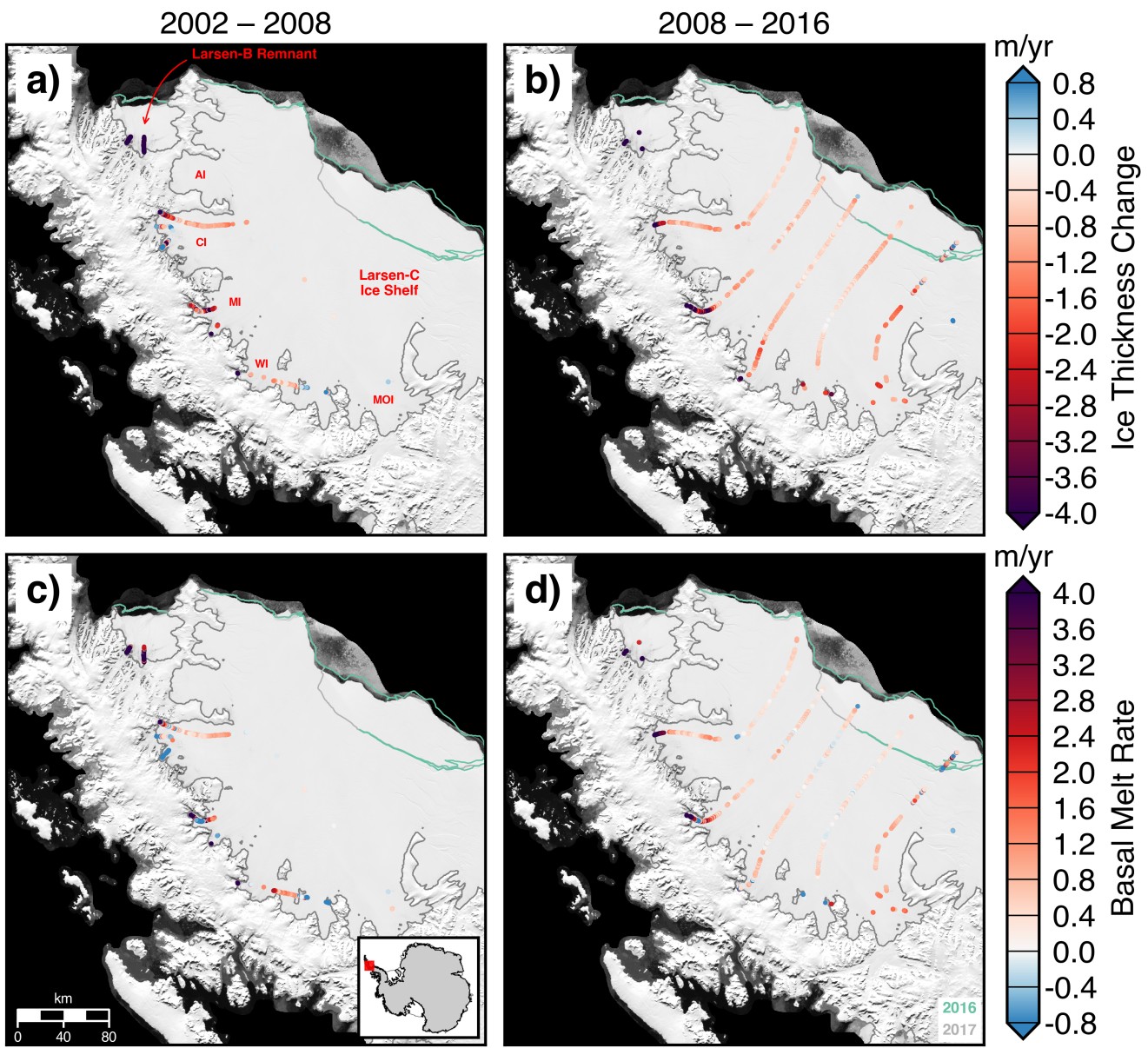

**Figure 6.** Ice thickness change (a-b) and estimated basal melt rates (c-d) of the Larsen-B remnant and Larsen-C Ice Shelf for two periods, 2002–2008 and 2008–2016. AI, CI, MI, WI and MOI denote the Adie, Cabinet, Mill, Whirlwind and Mobiloil Inlets, respectively. MEa-SUREs InSAR-derived Antarctic grounded ice boundaries are denoted in gray (Mouginot et al., 2017b). 2016 and 2017 ice shelf extents delineated from MODIS imagery are denoted in green and light gray, respectively (Scambos et al., 2001). Plots are overlaid on a 2008–2009 MODIS mosaic of Antarctica (Haran et al., 2014). Inset map denotes the location of the maps.

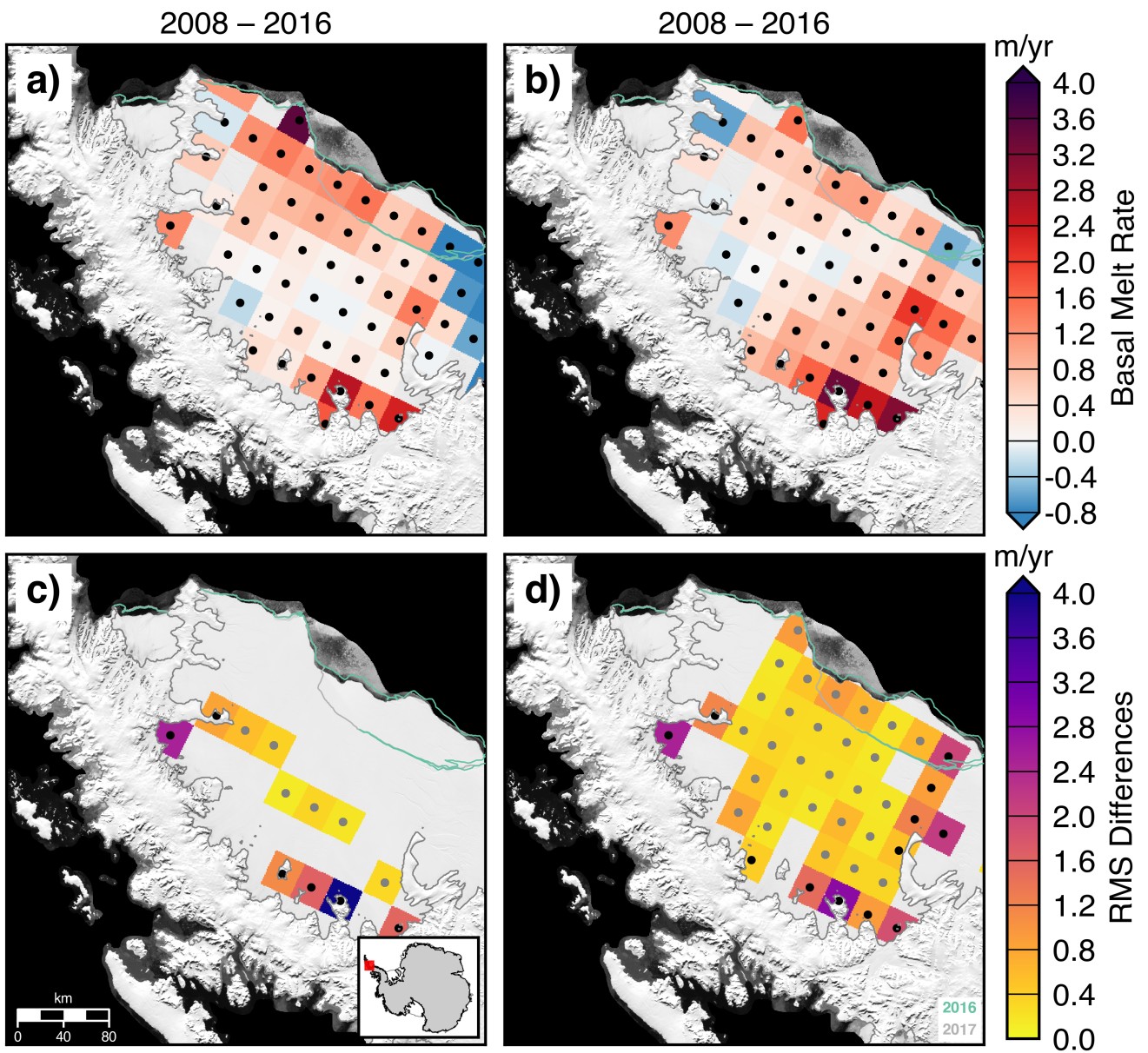

**Figure 7.** Estimated basal melt rates (a-b) from Adusumilli et al. (2018) and differences from melt rates derived from NASA/CECS Pre-IceBridge and NASA Operation IceBridge (c-d) of the Larsen-C Ice Shelf for two periods, 2002–2008 and 2008–2016. Stipples indicate locations with valid radar altimetry data (a-b) and coincident airborne laser altimetry data (c-d). MEaSUREs InSAR-derived Antarctic grounded ice boundaries are denoted in gray (Mouginot et al., 2017b). 2016 and 2017 ice shelf extents delineated from MODIS imagery are denoted in green and light gray, respectively (Scambos et al., 2001). Plots are overlaid on a 2008–2009 MODIS mosaic of Antarctica (Haran et al., 2014). Inset map denotes the location of the maps.

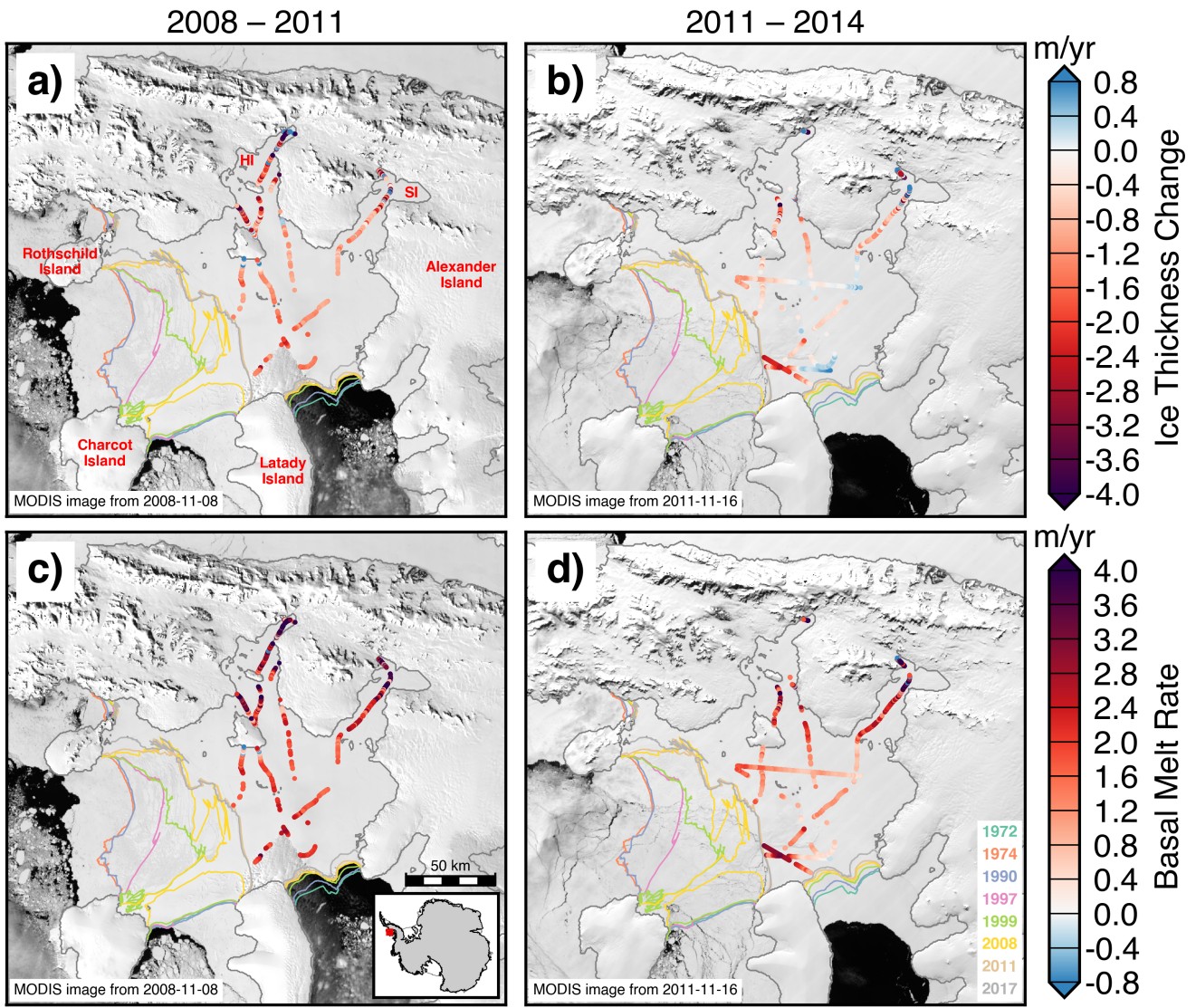

**Figure 8.** Ice thickness change (a-b) and estimated basal melt rates (c-d) of the Wilkins Ice Shelf for two periods, 2008–2011 and 2011–2014. HI and SI denote the Haydn and Schubert Inlets, respectively. MEaSUREs InSAR-derived Antarctic grounded ice boundaries are denoted in gray (Mouginot et al., 2017b). Historical ice shelf extents delineated from Landsat and MODIS imagery are denoted with colored lines. Plots are overlaid on MODIS images of Antarctic ice shelves provided by NSIDC (Scambos et al., 2001). Inset map denotes the location of the maps.

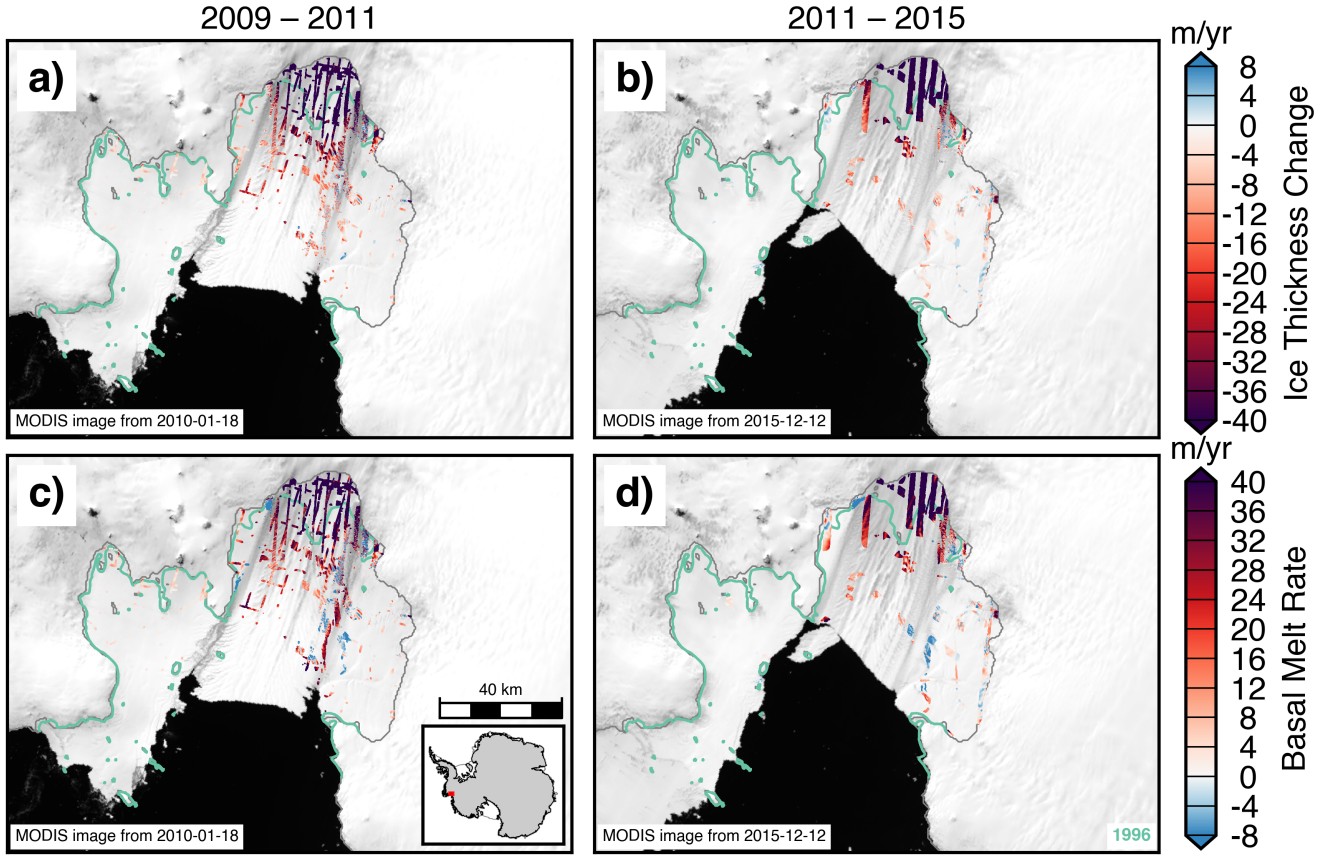

**Figure 9.** Ice thickness change (a-b) and estimated basal melt rates (c-d) of the Pine Island Ice Shelf for two periods, 2009–2011 and 2011–2015. MEaSUREs InSAR-derived Antarctic grounded ice boundaries are denoted in gray (Mouginot et al., 2017b). 1996 InSAR-derived grounding line locations from Rignot et al. (2016) are delineated in green. Plots are overlaid on MODIS images of Antarctic ice shelves provided by NSIDC (Scambos et al., 2001). Inset map denotes the location of the maps.

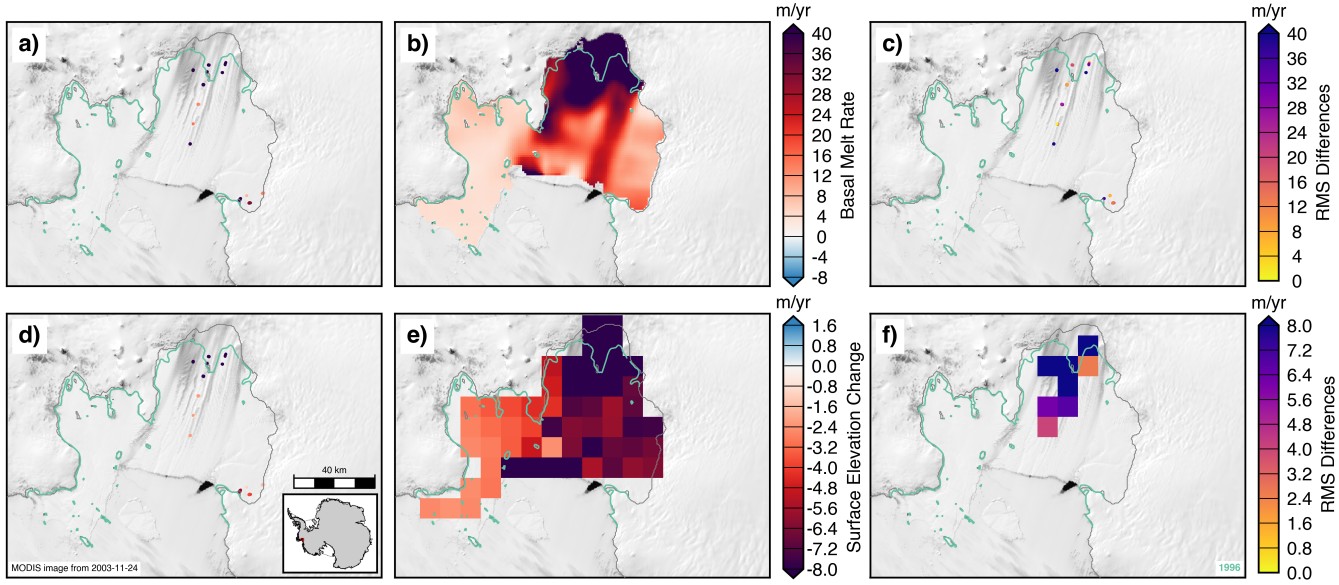

**Figure 10.** Estimated basal melt rates (a-c) and differences in melt rates (c) of the Pine Island Ice Shelf from (a) NASA/CECS Pre-IceBridge and NASA Operation IceBridge over 2002–2009 and (b) from ICESat over 2003–2009 (Rignot et al., 2013). Estimated elevation change rates (d-e) and differences in elevation change rates (f) of the Pine Island Ice Shelf from (d) NASA/CECS Pre-IceBridge and NASA Operation IceBridge over 2002–2009 and (e) from ICESat over 2003–2009 after correcting for strain (Pritchard et al., 2012) MEaSUREs InSAR-derived Antarctic grounded ice boundaries are denoted in gray (Mouginot et al., 2017b). 1996 InSAR-derived grounding line locations from Rignot et al. (2016) are delineated in green. Plots are overlaid on MODIS images of Antarctic ice shelves provided by NSIDC (Scambos et al., 2001). Inset map denotes the location of the maps.

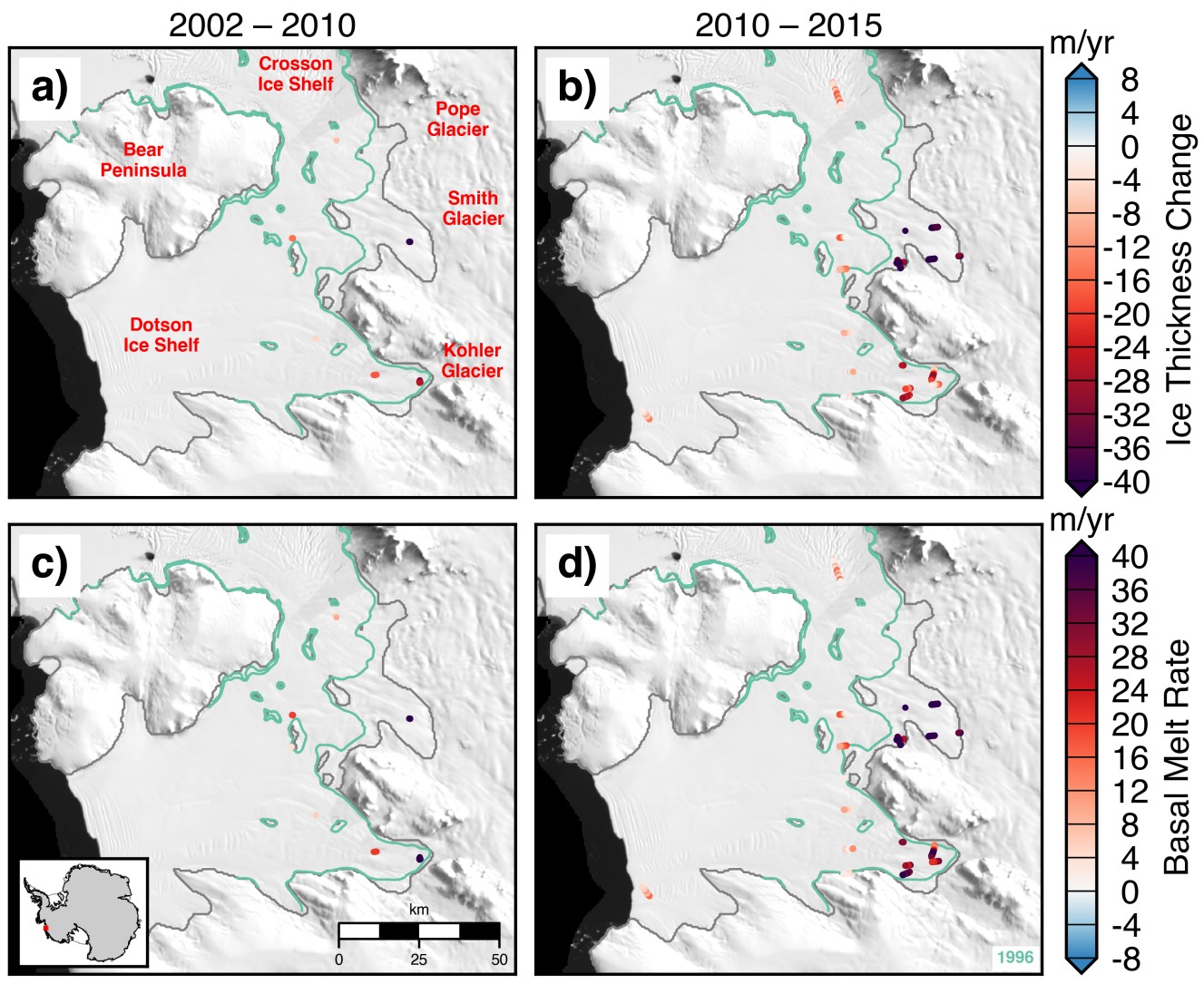

**Figure 11.** Ice thickness change (a-b) and estimated basal melt rates (c-d) of the Dotson and Crosson Ice Shelves for two periods, 2002–2010 and 2010–2015. MEaSUREs InSAR-derived Antarctic grounded ice boundaries are denoted in gray (Mouginot et al., 2017b). 1996 InSAR-derived grounding line locations from Rignot et al. (2016) are delineated in green. Plots are overlaid on a 2008–2009 MODIS mosaic of Antarctica (Haran et al., 2014). Inset map denotes the location of the maps.

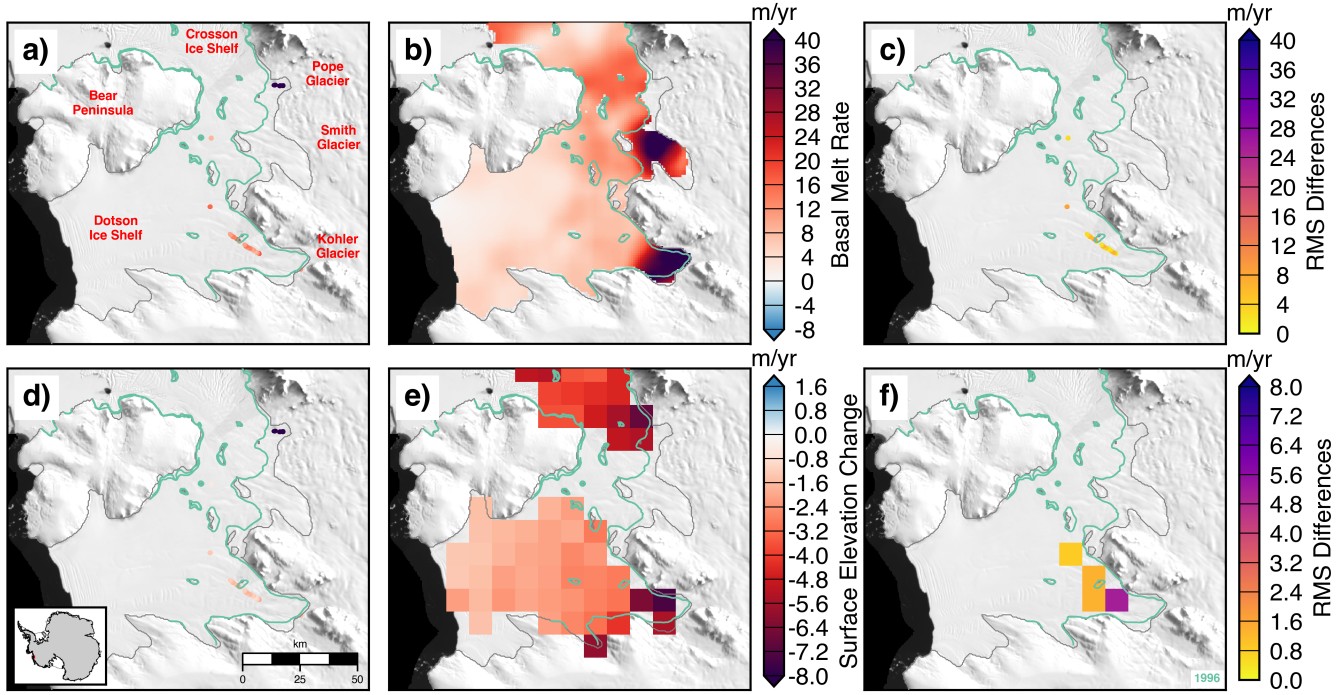

**Figure 12.** Estimated basal melt rates (a-b) and differences in melt rates (c) of the Dotson and Crosson Ice Shelves from (a) NASA/CECS Pre-IceBridge and NASA Operation IceBridge over 2002–2009 and (b) from ICESat over 2003–2009 (Rignot et al., 2013). Estimated elevation change rates (d-e) and differences in elevation change rates (f) of the Dotson and Crosson Ice Shelves from (d) NASA/CECS Pre-IceBridge and NASA Operation IceBridge over 2002–2009 and (e) from ICESat over 2003–2009 after correcting for strain (Pritchard et al., 2012). MEaSUREs InSAR-derived Antarctic grounded ice boundaries are denoted in gray (Mouginot et al., 2017b). 1996 InSAR-derived grounding line locations from Rignot et al. (2016) are delineated in green. Plots are overlaid on a 2008–2009 MODIS mosaic of Antarctica (Haran et al., 2014). Inset map denotes the location of the maps.

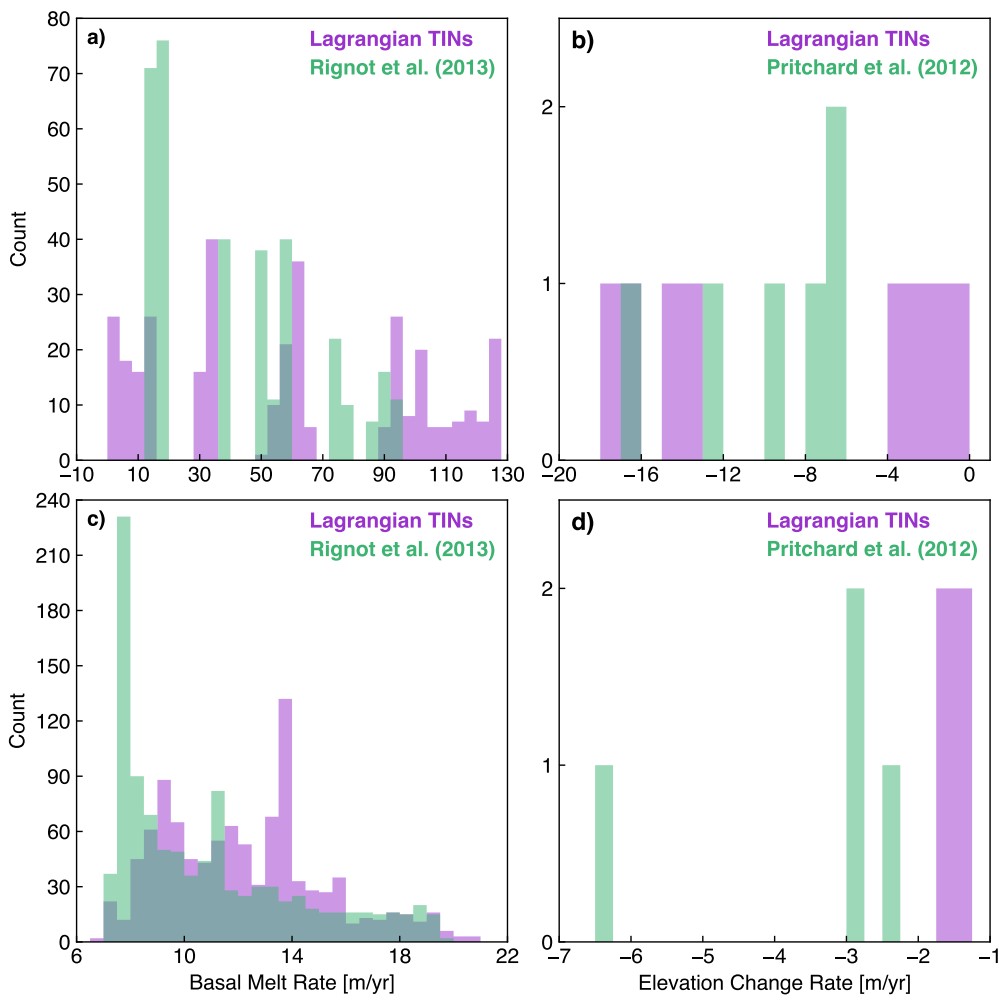

**Figure 13.** Histograms of basal melt rates (a,c) and surface elevation change (b,d) of the Pine Island Ice Shelf (a,b) and Dotson and Crosson Ice Shelves (c,d) from NASA/CECS Pre-IceBridge and NASA Operation IceBridge over 2002–2009 (purple) and from ICESat over 2003–2009 (green) using data from Rignot et al. (2013) and Pritchard et al. (2012).