# Peer review of "Antarctic Ice Shelf Thickness Change from Multi-Mission Lidar Mapping"

_The Cryosphere, 2018_

## Referee Comment (RC1) · Anonymous Referee #1 · 27 Dec 2018

This manuscript reports on estimates of thickness change and basal-melt rates along airborne survey lines over West Antarctic and Antarctic Peninsula ice shelves. These estimates were derived from lidar measurements (of surface height change) obtained from NASA's airborne campaigns between 2002 and 2015, combined with available surface velocity data from MEaSUREs/NSIDC, and surface-mass-balance and firn-state information from models (RACMO2.3, and a firn-densification model). The manuscript focuses on the methodology to invert height-change measurements from airborne lidar to basal-melt estimates in a Lagrangian framework. Finally, a brief discussion on the Lagrangian vs Eulerian approaches is presented, as well as putting in context some of the ice-shelf melt-rate values obtained.

I believe the results of this manuscript are of great value for comparing and calibrating

satellite-derived estimates of ice-shelf thickness change and melt rates. The authors put considerable effort to integrate all available/usable NASA airborne lidar data over the West Antarctic ice shelves. While these data set is quite sparse (only available along flight lines and with a few repeats), there are still very little data available to compare against the vast amount of satellite measurements, which makes this work of particular interest to the remote-sensing community. I have, however, several questions and suggestions that I would like to see addressed prior considering publication (see comments below).

Overall, the manuscript is well written and the figures are of good quality.

General comments:

I feel a thorough error assessment on derived melt-rate estimates is lacking. Given that, as mentioned in the manuscript itself, this set of estimates is expected to serve as a reference for published and future (e.g. from ICESat-2) satellite-derived estimates, I would expect a more comprehensive error assessment: How close to (available) in-situ measurements are these values? What are realistic confidence intervals given that some of the information comes from models? How sensitive are the estimated melt-rate values to unaccounted processes (due to lack of data or knowledge)?

Some of the short-time-scale (2 to 5 years) estimates are likely subject to the large interannual-to-decadal variability characteristic in the AS-BS sector (e.g. Paolo et al. 2015). For example, it has been shown that even ICESat-derived estimates (5-year period) can disagree substantially from longer-timescale averages (as those derived from radar altimetry). In many cases, the ICESat short time span (Prichard et all 2012; Rignot et al. 2013) overestimate the underlying decadal trend, simply because their estimates are focusing on the more variable short-term scales.

Substantial (and important) information on the methodology is being introduced in the Discussion section of the manuscript. I understood some aspects/limitations of the methodology only after reaching the discussion page (which is the final portion of the

[Figure]

Main text).

Can direct comparisons with previously published estimates be made for some locations (using, for example, Pritchard et al. 2012; Rignot et al. 2013; Paolo et al. 2015; Adusumilli et al. 2018)? This would be very valuable and could motivate good discussion regarding discrepancies and/or similarities.

Specific comments:

p2, l3-4: "accelerated 2 to 8 times their previous flow rates"... Please define "previous", i.e., when those measurements were taken (right before 2002, or five/ten years before)?

p2, l5: "surface thinning"... Are you referring to thinning of the firn layer (i.e. densification), which I don't think any of the provided references support this? Or perhaps you mean "surface lowering"?

p2, l7: What is an "internal change in ice dynamics" (as opposed to "an external change")?

p2, l8: ocean melt -> ocean-driven melt

p2, l25: "over Pre-IceBridge and NASA Operation IceBridge campaigns is shown"... Do you mean "prior to and during NASA Operation IceBridge campaigns is shown..."?

p2, l27-28: What exactly do the 'converted' heights represent? Height w.r.t. an ellipsoid model or w.r.t. a geoid model... it seems you are tracking deviations from the geoid, and why you need this conversion? Perhaps to invert for thickness/basal melt, but it is not clear at this point in the text.

p3, l7: What is "the scale of the individual triangular facet"?

p3: On "Tidal and Non-Tidal Ocean Variation":

Armitage et al. (2018) showed substantial sea-level anomalies (changes w.r.t. mean

sea level) around Antarctica: about 3 cm at seasonal scales and 5 cm associated with the ENSO cycle. How will these translate to/impact the derived ice-shelf height changes? At the very least, these should be accounted for in the error budget. Note that these SLAs around Antarctica could not be precisely measured until only recently (e.g. Armitage et al).

What precisely are the "Non-tidal sea surface anomalies over ice-free ocean points", i.e., what process are you removing with the CMEMS product? Is this accounting for spatially variable sea-level rise? For example, Paolo et al. (2015) corrected for rates of sea-level change around Antarctica varying from 2 to 4 mm/yr (compared to the global mean of ∼3 mm/yr)

p4, l8-10: What's the relevance of "highly complex topography of mountains and glacial valleys" if you are working over (relatively flat) ice shelves? I'm saying this because I haven't seen a comparison between the 27km and 5.5km SMB models against in-situ measurements specifically *over* the ice shelves, to be convinced that the higher-res product does provide a more accurate representation of SMB state over flat surfaces.

p4, l13-14: I'm confused here: "The absolute precision of the RACMO2.3p2 model outputs has been estimated...", are you referring to the latest high-res model (the 5.5 km)? If so, why is the reference from 2006 (I assume they did not have the high-res model back then)?

p5, l3-4: What is "basal thickness change rate"? Changes in ice-shelf thickness due to mass loss/gain at the bottom? Or...

Fig 10: "The elevation change rates shown here are not corrected for oceanic or surface processes and are not RDE filtered"... Why not?

General comment: I don't know what 'basal thickness change means'... Thickness change solely due to basal mass change? Please be more specific/accurate.

p5, l33-35: Could the difference in melt rate near the grounding zone be explained

simply by the (large) interannual-to-decadal variability in the AS sector (as shown, for example, by Dutrieux et al. 2014; Paolo et al. 2015; Jenkins et al, 2018)?

p6, l15-16: However, Lagrangian estimates miss the grounding lines due to the direction of ice flow from grounded to floating. That is, sampled sites near the grounding lines were previously over grounded ice, lacking the corresponding measurement pair for comparison. This limitation affects measurements downstream of the GL depending on time separation between data points and flow speed. Another limitation of the Lagrangian approach is the sparseness of the estimates (compared to Eulerian solutions) since not all measurements will have a matching upstream pair (as also demonstrated by Moholdt et al. 2014). Further, in the case of airborne surveys where the flight segments do not cross the entire ice shelf, measurements on the downstream end of the transect will also lack corresponding matching pairs.

p6, l18-20: Substantial smoothing was required because the effect of ice advection and divergence was not corrected for. With high-quality velocity products available today (e.g. Rignot et al. 2017, NSIDC; Gardner et al. 2018) the flux-divergence signal can and should be removed (or at least reduced substantially) from the basal mass balance estimates (see for example, Berger et al. 2017; Lilien et al. 2018; Adusumilli et al. 2018).

p6, l19-20: "spatial smoothing [...] to filter out the effects of advection"... This misleading. The smoothing is not targeting specifically the advection-related features, instead, is removing everything that falls within the cutoff frequency of the smoother.

p6, l32-33: I think a more comprehensive "update" (to Pritchard et al. 2012) has already been presented (see Paolo et al. 2015)... or not?

p7, first para: The discussion on the limited velocity coverage back in time for Lagrangian estimation is important (modern Eulerian estimates also depend on the removal of the advection signal). I feel the authors should go beyond just discussing and try and quantify the effect (i.e. the contribution to the error budget) of potential changes

in ice flow. In other words, how sensitive are the melt rate estimates to changing velocity magnitudes? Typical magnitudes of velocity change can be taken from the literature for the few locations they are available (e.g. Mouginot et al. 2014).

---

## Referee Comment (RC2) · Anonymous Referee #2 · 3 Jan 2019

SUMMARY

The authors use airborne laser altimetry (from airborne topographic mappers (ATM)) over Antarctic Peninsula (AP) and Amundsen Sea (AS) ice shelves, plus models of surface mass balance and firn compaction, to measure ice shelf thinning rates and assign these rates to individual terms in the mass balance.

The study is complementary to several previous studies that used satellite altimeters. The coverage of ATM is poor prior to Operation Icebridge (OIB). However, it has some advantages in terms of dedicated tracks, in particular allowing measurements to get close to grounding lines. It is therefore a valuable study, and dataset, to provide to the community.

[Figure]

Note that I have read the comments by Anonymous Referee #1 and agree with most of those, which I won't generally repeat. It seems unlikely that regional sea level trends could matter much, although it could amount to ~5 cm of elevation in a decade. It's possible that general ocean variability that isn't corrected for (currently only tides and inverse barometer) is a bigger source of error especially given that the ATM missions are essentially instantaneous, and sparse in time.

GENERAL

1. I spent a lot of the paper being confused by the term "ice thickness change rates". This relates to the use of Lagrangian calculations. The authors explain why they use Lagrangian methods, which makes sense, although it often seems to lead to massive data loss: compare figure 1 flight lines with locations of ice thickness change on figures 5, 7, 8 and 9. However, Lagrangian methods are really just a tool to get the mass balance terms. The most important thing is whether the ice shelf is losing mass, and the spatial distribution of that loss, so that Eulerian variability is really what you want to report in terms of SMB, BMB and divergence.

If you agree with that, then the important "ice thickness change rate" is Eulerian, which you get back from Lagrangian by adding back in the strain thinning and advection terms. (If they appear to be changing, that's relevant too.) The simplest approach to clarify what you're reporting would be to introduce Eulerian and Lagrangian rate symbols early (d/dt and D/Dt), then use the symbol rather than the words. Every time I see capital 'D', I'll know it is Lagrangian.

2) It is strange that Results are presented first, then back to Methods, as far as figures go. Given how much the data distribution thins out from Fig. 1 to the Lagrangian maps, the first thing to do would be to determine if Lagrangian is a good method. Potentially, you are better off with averaging of a lot of noisy Eulerian measurements rather than far fewer cleaner Lagrangian values. I'd move Fig. 6 to Fig. 5, demonstrating the value of along-flowline repeat ATM, then next I'd have something like Figure 10 to make

points about the value of Lagrangian vs Eulerian. You need to also check that you are comparing the same things here: results from Eulerian TINs should be the same average as Lagrangian TINS provided the Lagrangian values have been re-corrected for advection and strain.

3) The authors should look at another Cryosphere Discussions paper by Shean et al. (https://www.the-cryosphere-discuss.net/tc-2018-209/), where Pine Island melt rates are assessed using high-res image-based Lagrangian processing.

4) The authors should probably compare their results for Larsen C with the ATM measurements presented in Adusumilli et al. (2018: GRL).

5) Overall, I think this paper fails to exploit the key features of ATM vs satellite-based products. Satellite altimeters and stereo imagery (Shean et al., and an earlier Dutrieux et al (I think) paper), make the process easier, but all satellite altimeters lack spatial resolution and radar altimeters struggle near grounding lines and other steep regions. Think about the new science that is available from a carefully compiled ATM data set where all the biases have been corrected for. If there is no new science, the data set is still valuable as it provides independent estimates to compare with the satellite-derived values. In this case, the most obvious value of the data set is as intended for OIB: a continuation of the ICESat laser altimeter record. Why not look at ICESat data as a third, earlier period in the various plots that compare pre-2011 and post-2011 data?

MAJOR: SPECIFIC

p.1/l.2: See general comments. The reader needs to know whether you mean Lagrangian or Eulerian ice thickness change, and if the Lagrangian estimates have been re-corrected back to Eulerian.

p.1/l.8-9: Comments on Larsen C depend on the quality of the SMB and firn models. This sentence suggests that the ice thickness change really is DH/dt, not dh/dt.

p.1/l.9-11: I don't think *you* show that Wilkins depends on "short time-scale and

upper-ocean processes": the only evidence I see for this is citations to previous work.

p.1/l.11-12: Again, this is where you'd be better off reporting dH/dt, even if you're deriving it via re-corrected DH/Dt. I was surprised that PIG was "thinning" by 40 m/yr, even close to the grounding line. The more important numbers are in comparisons: you want to show actual Eulerian thinning (dH/dt), BMB, and maybe the ice divergence term.

p.2/l.31: The Shean et al. TCD paper is another example of Lagrangian processing.

p.3/l.27-29: I don't understand how you remove non-tidal ocean height change for ice-free ocean points from ice-shelf data. Extrapolate under the ice front? Do you get AVISO sea surface height all the way to the ice fronts at all times of ATM surveys, or does sea ice get in the way? What processes do you think the AVISO products are correcting for, or is this a coarse approximation for regional sea level rise?

p.4/l.20 ff: You need to explain all the terms in this equation immediately.

p.5-6 (Results): This would be clearer if you used sub-headers for each ice shelf that you are considering: Larsen C, Wilkins, Pine Island, and Dotson/Crosson. Also, this is a critical place to use symbols regarding ice thickness change: is it Eulerian dH/dt, Lagrangian DH/Dt, or Lagrangian-derived Eulerian dh/dt?

p.5/l.27-28: Sentence starting "These periods" suggest that RACMO2.3p2 ASE055 is only available for specific periods, which then determine the breakdown of ATM into different epochs. Is this true? Regardless, the reader needs to know the period for which this high-res surface processes model is available.

p.5/l.32-33: Rignot and Jacobs (2002) is not the right cite for "highest impact on glacial flow dynamics". They just assume that and use it to justify looking at melt rates near the grounding line. There are many more recent papers that might be relevant, e.g., Walker et al. (2008), Gagliardini et al. (2010), probably others.

p.6/l.3: Rignot (2002) seems like a strange single choice for citation here.

p.6/l.4-13: The Dotson/Crosson data are incredibly sparse, which I assume is a consequence of using Lagrangian processing given data density on Fig. 1. So (a) is this a place where higher noise in Eulerian would have been better? (b) Maybe you haven't enough data to learn whether conditions are different from the ICESat-era results of Khazendar et al. 2016? This points again to using the ICESat-era results as a natural comparison for the more recent ATM.

p.6/l.17: The statement "Our Eulerian approach" seems to contradict everything you've said about using a Lagrangian approach. This comparison should be much earlier in "Methods", then you could mention "We began by calculating …. using three approaches, …, …, and …, applied to Larsen C. Results (Fig. X) demonstrate that …." Just make sure the figure really does compare Eulerian with re-corrected Lagrangian, or advection-and-strain-corrected Eulerian with Lagrangian.

p.6/l.26-27: Dotson/Crosson data are extremely sparse, and it isn't at all clear that Lagrangian methods are the best approach here.

p.6/l.29-30: the statement "would likely not be representative" is probably true, especially for Dotson/Crosson, but needs to be justified, e.g., on the basis of data sparseness.

p.6/l.30-32: It isn't really obvious why you need a DEM, specifically from photogrammetry, to use ICESat-2 for dH/dt. It helps with the advection terms and Lagrangian TINs, but maybe you need to set up the idea better, along the lines of "The Lagrangian method is strongly dependent on a detailed understanding of surface topographic features being advected by the ice flow …"

p.7/l.13-22: (a) This section does not flow well, but it does raise two issues that you haven't really explained well up to now:

(a) Pros and Cons of radar vs laser. The goal, probably, is change in vertically integrated *mass* (or ice-equivalent thickness). With laser, you get true surface height

really well, but conversion to mass depends on the firn model. If the snow layer is lighter than you thought, you infer too much mass. With radar, it is complicated by penetration (and footprint size), but on the other hand maybe that's good because the inferred reflecting surface is below the lightest snow. However, you still need the model of firn compaction below the reflector.

(b) The study hasn't really been set up as well as it could have. This gets back to: Is there really new scientific insight here, or is the goal mainly to provide an independent data set that is of specific value in comparing with satellite-derived ice-shelf changes, specifically laser-based? Either way is good for a paper, with the latter being the justification for OIB anyway. A clearer goal, stated early, might help organize the paper so that results are written around that goal. At the moment the paper reads like you're identifying new science, but the Results section mainly relies on, or repeats, previous studies, just with a new data set. e.g., Adusumilli et al. (2018) reach the same conclusions regarding Larsen C, except they don't spend much time of the advection-and-strain terms, but they do use ATM as validation. Wilkins is interesting, but why not compare ATM tracks with ICESat to get a better sense of pre- and post-ICESat behavior?

TECHNICAL: SPECIFIC

p.1/l.19-20: (a) I think Rignot et al. 2013 just assumes that ice shelves buttress grounded ice, don't show it. You can't cite every paper that makes that claim. (b) Sentence starting "The thinning . . ." just repeats the idea of buttressing.

p.2/l.1-2: Again, you're repeating the buttressing argument.

p.2/l.19: Abbreviation "WFF" isn't used again, so not needed.

p.2/l.32: Here you cite Rignot et al 2017 for MEASURES, but on p.4/l.24 it is Mouginot et al. 2017a.

p.4/l.28: This reads like the range of validity for hydrostatic is only the narrow band of

1-8 km from the grounding line. You mean that this region is *not* hydrostatic, but that the flexural boundary width is in this range.

General style, especially in Results: You make a habit of starting paragraphs with "Figure X shows ...". This sounds like you have a collection of figures to describe, rather than making figures to fit your narrative.

General Style: "{Name} ice shelf" or "{Name} Ice Shelf" ?

p.6/l.2: Why refer to Fig. 8*b*, specifically?

p.6/l.5: I think this means "two periods – 2002-2010 and 2010-2015 – are shown in"

p.7/l.8: more precisely "maps of time-varying velocity"

p.7/l.9-12: Would be good to have cites to each of these products.

---

## Author Response (AR1)

Dear Kenny Matsuoka,

We are very appreciative for the reviews of our manuscript "Antarctic Ice Shelf Thickness Change from Multi-Mission Lidar Mapping." In response to the reviewer comments, we have revised the manuscript to clarify some essential points and add a comparative analysis with Adusumilli et al. (2018). The modifications did not change the overall conclusions or results.

In the revision, we include:

- 1. A point-by-point response to the reviewer comments. Responses are *italicized and gray*.
- 2. An enumerated list of the modifications made to the manuscript.
- 3. A copy of the manuscript with the changes noted. (Highlighted with red struck-through text to denote subtractions and blue underlined text to denote additions)
- 4. A final copy of the manuscript with those same changes incorporated.

Regards, Tyler C. Sutterley

**First Reviewer Comments:**

This manuscript reports on estimates of thickness change and basal-melt rates along airborne survey lines over West Antarctic and Antarctic Peninsula ice shelves. These estimates were derived from lidar measurements (of surface height change) obtained from NASA's airborne campaigns between 2002 and 2015, combined with available surface velocity data from MEaSUREs/NSIDC, and surfacemass-balance and firn state information from models (RACMO2.3, and a firn-densification model). The manuscript focuses on the methodology to invert height-change measurements from airborne lidar to basal-melt estimates in a Lagrangian framework. Finally, a brief discussion on the Lagrangian vs Eulerian approaches is presented, as well as putting in context some of the ice-shelf melt-rate values obtained. I believe the results of this manuscript are of great value for comparing and calibrating satellite-derived estimates of ice-shelf thickness change and melt rates. The authors put considerable effort to integrate all available/usable NASA airborne lidar data over the West Antarctic ice shelves. While these data set is guite sparse (only available along flight lines and with a few repeats), there are still very little data available to compare against the vast amount of satellite measurements, which makes this work of particular interest to the remote-sensing community. I have, however, several questions and suggestions that I would like to see addressed prior considering publication (see comments below). Overall, the manuscript is well written and the figures are of good quality.

Thank you. We appreciate the thoroughly beneficial review of our manuscript. We address your comments point-by-point and update the manuscript accordingly.

**General comments:**

I feel a thorough error assessment on derived melt-rate estimates is lacking. Given that, as mentioned in the manuscript itself, this set of estimates is expected to serve as a reference for published and future (e.g. from ICESat-2) satellite-derived estimates, I would expect a more comprehensive error assessment: How close to (available) in-situ measurements are these values?

We expanded upon our error calculation in the manuscript. There are tide gauges around Antarctica that are used for validating the CATS2008 model. The 5.5km SMB models are validated against Operation IceBridge snow radar observations, satellite melt observations, and and in-situ observations (Kuipers Munneke et al., 2017; Lenaerts et al., 2018).

**What are realistic confidence intervals given that some of the information comes from models?**

This is an excellent question. We assume a 15% uncertainty in surface mass balance and firn height following estimates from Kuipers Munneke et al. (2017). Tidal uncertainties are estimated using the constituent RMS values from King et al. (2011). Uncertainties in flux divergence were estimated using annually resolved velocity maps (Mouginot et al., 2017a) and uncertainties in Bedmap2 ice thickness (Fretwell et al., 2013).

**How sensitive are the estimated melt-rate values to unaccounted processes (due to lack of data or knowledge)?**

Great question. Because the ice shelves are largely in hydrostatic equilibrium, any uncertainty in terms of elevation will be magnified by approximately  $10 \times$  in the final estimates of thickness change and basal melt rate.

Some of the short-time-scale (2 to 5 years) estimates are likely subject to the large interannual-todecadal variability characteristic in the AS-BS sector (e.g. Paolo et al. (2015)). For example, it has been shown that even ICESat-derived estimates (5-year period) can disagree substantially from longertimescale averages (as those derived from radar altimetry). In many cases, the ICESat short time span (Pritchard et al. (2012); Rignot et al. (2013)) overestimate the underlying decadal trend, simply because

**their estimates are focusing on the more variable short-term scales.**

Absolutely. At present, records from laser altimetry are far less compete than records from radar altimetry in terms of temporal resolution and duration (Paolo et al., 2015; Adusumilli et al., 2018). However, laser altimetry datasets have more accurate surface determination and can more accurately track over regions of abrupt topographical change. ICESat-2 should provide a valuable extension to the laser altimetry record and help separate short-term oscillations with long-term change.

Substantial (and important) information on the methodology is being introduced in the Discussion section of the manuscript. I understood some aspects/limitations of the methodology only after reaching the discussion page (which is the final portion of the Main text).

Good point. Text and figures have been reordered for improved continuity and presentation.

Can direct comparisons with previously published estimates be made for some locations (using, for example, Pritchard et al. (2012); Rignot et al. (2013); Paolo et al. (2015) and Adusumilli et al. (2018)? This would be very valuable and could motivate good discussion regarding discrepancies and/or similarities.

Great point. We have added a direct comparison with the results from Adusumilli et al. (2018) and have added points emphasizing this purpose of our dataset. We did not compare with data from Paolo et al. (2015) as the publicly available data is for a different time period. We did not compare with Pritchard et al. (2012) as the data is not provided in a compiled form. Rignot et al. (2013) do not provide publicly available data.

**Specific comments:**

**p2, I3-4:** "accelerated 2 to 8 times their previous flow rates"... Please define "previous", i.e., when those measurements were taken (right before 2002, or five/ten years before)?

Great point. Added that the before and after measurements were taken in 1996 and 2003. "In 2003, a year after the collapse of the Larsen B ice shelf, some tributary glaciers draining into the Weddell Sea from the Antarctic Peninsula were flowing 2–8 times their 1996 flow rates (Rignot et al., 2004). These glaciers continued flowing at accelerated rates years after the collapse (Rignot et al., 2008; Berthier et al., 2012)."

p2, I5: "surface thinning"... Are you referring to thinning of the firn layer (i.e. densification), which I don't think any of the provided references support this? Or perhaps you mean "surface lowering"?

Clarified to mean "dynamic thinning" as noted in Pritchard et al. (2009) and Flament and Rémy (2012).

**p2, I7: What is an "internal change in ice dynamics" (as opposed to "an external change")?**

Changed to "The dynamical change of these glaciers..."

**p2, I8:** ocean melt  $\rightarrow$  ocean-driven melt

Done. Thank you.

**p2, I25:** "over Pre-IceBridge and NASA Operation IceBridge campaigns is shown"... Do you mean "prior to and during NASA Operation IceBridge campaigns is shown..."?**

Changed to "The spatial coverages of each instrument in Antarctica for the campaigns prior to and during NASA Operation IceBridge are shown in Figure 1." Thank you.

p2, I27-28: What exactly do the 'converted' heights represent? Height w.r.t. an ellipsoid model or w.r.t. a geoid model...it seems you are tracking deviations from the geoid, and why you need this conversion? Perhaps to invert for thickness/basal melt, but it is not clear at this point in the text.

Good point. Changed to "In order to track changes in ice shelf freeboard, the ellipsoid heights for each instrument were then converted to be in reference to the GGM05 geoid using gravity model coefficients provided by the Center for Space Research (Ries et al., 2016)."

**p3, I7: What is "the scale of the individual triangular facet"?**

Added that an individual triangle is  $\sim$  10–100m.

p3: On "Tidal and Non-Tidal Ocean Variation": Armitage et al. (2018) showed substantial sea-level anomalies (changes w.r.t. mean sea level) around Antarctica: about 3 cm at seasonal scales and 5 cm associated with the ENSO cycle. How will these translate to/impact the derived ice-shelf height changes? At the very least, these should be accounted for in the error budget. Note that these SLAs around Antarctica could not be precisely measured until only recently (e.g. Armitage et al. (2018)). What precisely are the "Non-tidal sea surface anomalies over ice-free ocean points", i.e., what process are you removing with the CMEMS product? Is this accounting for spatially variable sea-level rise? For example, Paolo et al. (2015) corrected for rates of sea-level change around Antarctica varying from 2 to 4 mm/yr (compared to the global mean of ~3 mm/yr)

Good point. Paolo et al. (2015) used the same dataset from AVISO in their study (described in their supplementary materials). We clarify that the sea surface anomalies removed are local sea level change, which will include long-term sea level rise and inter-annual fluctuations. We add the sentence "Regional sea levels fluctuate due to changes in ocean dynamics, ocean mass, and ocean heat content (Church et al., 2011; Armitage et al., 2018)." We also include that the sea surface anomalies are added to estimates of mean dynamic topography, which are the mean deviations of the sea surface from the geoid. "The non-tidal sea surface anomalies are added to estimates of mean deviation of the sea surface from the Earth's geoid due to ocean circulation."

**p4, I8-10:** What's the relevance of "highly complex topography of mountains and glacial valleys" if you are working over (relatively flat) ice shelves? I'm saying this because I haven't seen a comparison between the 27km and 5.5km SMB models against in-situ measurements specifically \*over\* the ice shelves, to be convinced that the higher-res product does provide a more accurate representation of SMB state over flat surfaces.

Fair point. While there is little difference for the ice shelves in the Amundsen Sea, there are some substantial differences for the ice shelves in the Weddell Sea. The major difference is how well the topography of the peninsula is resolved at 5.5km versus 27km. Resolving some downstream effects within the climate model requires the highest-possible spatial resolution topography (Datta et al., 2018). Added "The higher spatial resolution topography improves the modeling of wind-driven downstream effects over ice shelves (Datta et al., 2018)."

p4, I13-14: I'm confused here: "The absolute precision of the RACMO2.3p2 model outputs has been estimated...", are you referring to the latest high-res model (the 5.5 km)? If so, why is the reference from 2006 (I assume they did not have the high-res model back then)?

The van de Berg et al. (2006) citation was for the method used for evaluation of the RACMO2 models. We updated the sentence to say that it is "following Kuipers Munneke et al. (2017) and Lenaerts et al. (2018)".

**p5, I3-4:** What is "basal thickness change rate"? Changes in ice-shelf thickness due to mass loss/gain at the bottom? Or...**

*Correct. It referred to changes in ice-shelf thickness due to losses at the base. We changed the maps to use two separate colorbars and show basal melt rates in meters of ice per year.*

**Fig 10: "The elevation change rates shown here are not corrected for oceanic or surface processes and are not RDE filtered"... Why not?**

Fair question. The original intent was to only show the differences due to the processing method. We updated the figure to correct for ocean and surface processes and we noted that we have corrected for strain in the Eulerian-derived values. The data isn't RDE filtered in order to show the worst case of each technique (such as near the rifting that developed before the calving of the A-68 iceberg).

**General comment:** I don't know what 'basal thickness change means'... Thickness change solely due to basal mass change? Please be more specific/accurate.**

Yes, this is what it referred to in the previous manuscript. All plots have been changed to show basal melt rates in terms of meters of ice per year.

**p5, I33-35:** Could the difference in melt rate near the grounding zone be explained simply by the (large) interannual-to-decadal variability in the AS sector (as shown, for example, by Dutrieux et al. (2014); Paolo et al. (2015); Jenkins et al. (2018))?

Yes. This point has been added to the text.

p6, I15-16: However, Lagrangian estimates miss the grounding lines due to the direction of ice flow from grounded to floating. That is, sampled sites near the grounding lines were previously over grounded ice, lacking the corresponding measurement pair for comparison. This limitation affects measurements downstream of the GL depending on time separation between data points and flow speed. Another limitation of the Lagrangian approach is the sparseness of the estimates (compared to Eulerian solutions) since not all measurements will have a matching upstream pair (as also demonstrated by Moholdt et al. (2014)). Further, in the case of airborne surveys where the flight segments do not cross the entire ice shelf, measurements on the downstream end of the transect will also lack corresponding matching pairs.

These are great points that we have been added to the methods and discussions sections.

" In order to minimize the possibility of co-registering measurements over ice shelves with measurements over grounded ice near the grounding zone or measurements over the ocean, sea ice floes and icebergs, we only include points that are on the ice shelf for both time periods using grounded ice delineations from Rignot et al. (2016) and Mouginot et al. (2017b) and ice shelf extents manually digitized from Landsat (LPDAAC) and MODIS imagery (Scambos et al., 2001)."

**p6, I18-20:** Substantial smoothing was required because the effect of ice advection and divergence was not corrected for. With high-quality velocity products available today (e.g. Rignot et al. (2017); Gardner et al. (2018)) the flux-divergence signal can and should be removed (or at least reduced substantially) from the basal mass balance estimates (see for example, Berger et al. (2017); Lilien et al. (2018); Adusumilli et al. (2018)).

Excellent point. We have noted this in the text.

**p6**, **I19-20**: "spatial smoothing [...] to filter out the effects of advection"... This misleading. The smoothing is not targeting specifically the advection-related features, instead, is removing everything that falls within the cutoff frequency of the smoother.

Fair point. While one of the main noise sources for ice shelves are these advected features, it is absolutely correct that the filters were not specifically used to remove these artifacts. This portion has been removed.

**p6, I32-33:** I think a more comprehensive "update" (to Pritchard et al. (2012)) has already been presented (see Paolo et al. (2015))... or not?

Fair point. While they are based on different datasets (radar vs. laser), Paolo et al. could be considered an update to Pritchard et al.. This sentence has been removed.

p7, first para: The discussion on the limited velocity coverage back in time for Lagrangian estimation is important (modern Eulerian estimates also depend on the removal of the advection signal). I feel the authors should go beyond just discussing and try and quantify the effect (i.e. the contribution to the error budget) of potential changes in ice flow. In other words, how sensitive are the melt rate estimates to changing velocity magnitudes? Typical magnitudes of velocity change can be taken from the literature for the few locations they are available (e.g. Mouginot et al. (2014)).

This is an excellent point. We include estimates of annual changes in flux divergence in our error budgets. Including time-variable velocity maps to advect the locations of the elevation measurements in our Lagrangian methodologies is the subject of future work.

**Second Reviewer Comments**

**SUMMARY**

The authors use airborne laser altimetry (from airborne topographic mappers (ATM)) over Antarctic Peninsula (AP) and Amundsen Sea (AS) ice shelves, plus models of surface mass balance and firn compaction, to measure ice shelf thinning rates and assign these rates to individual terms in the mass balance. The study is complementary to several previous studies that used satellite altimeters. The coverage of ATM is poor prior to Operation Icebridge (OIB). However, it has some advantages in terms of dedicated tracks, in particular allowing measurements to get close to grounding lines. It is therefore a valuable study, and dataset, to provide to the community.

Note that I have read the comments by Anonymous Referee #1 and agree with most of inverse barometer) is a bigger source of error especially given that the ATM missions are essentially instantaneous, and sparse in time.

We are really appreciative for the helpful review of our manuscript. In response, we have revised the manuscript and clarified some essential points.

We completely agree that other sources of oceanic variability can influence the measurements over ice shelves. In the current and previous versions of the manuscript, sea level variations are accounted for using AVISO products distributed by Copernicus. The use of these sea level anomaly products follows the ice shelf work of Paolo et al. (2015).

**GENERAL**

1. I spent a lot of the paper being confused by the term "ice thickness change rates". This relates to the use of Lagrangian calculations. The authors explain why they use Lagrangian methods, which makes sense, although it often seems to lead to massive data loss: compare figure 1 flight lines with locations of ice thickness change on figures 5, 7, 8 and 9. However, Lagrangian methods are really just a tool to get the mass balance terms. The most important thing is whether the ice shelf is losing mass, and the spatial distribution of that loss, so that Eulerian variability is really what you want to report in terms of SMB, BMB and divergence.

Figures and text have been changed to use basal melt rates (in terms of meters of ice equivalent per year). The data is spatially sparse over ice shelves regardless of reference frame (especially in the pre-IceBridge era). After 2009, it is possible to have nearly annually resolved estimates of ice thickness change along the flight lines for some ice shelves. Idealistically, reporting Eulerian variability would be preferable over Lagrangian variability. However, substantial smoothing or averaging is required with Eulerian-derived estimates to reduce the impact of noise, and thus Lagrangian-derived estimates can provide more accurate solutions if the spatial coverage isn't comprehensive.

If you agree with that, then the important "ice thickness change rate" is Eulerian, which you get back from Lagrangian by adding back in the strain thinning and advection terms. (If they appear to be changing, that's relevant too.) The simplest approach to clarify what you're reporting would be to introduce Eulerian and Lagrangian rate symbols early (d/dt and D/Dt), then use the symbol rather than the words. Every time I see capital 'D', I'll know it is Lagrangian.

**Nomenclature has been updated.**

2. It is strange that Results are presented first, then back to Methods, as far as figures go. Given how much the data distribution thins out from Fig. 1 to the Lagrangian maps, the first thing to do would

be to determine if Lagrangian is a good method. Potentially, you are better off with averaging of a lot of noisy Eulerian measurements rather than far fewer cleaner Lagrangian values. I'd move Fig. 6 to Fig. 5, demonstrating the value of along-flowline repeat ATM, then next I'd have something like Figure 10 to make points about the value of Lagrangian vs Eulerian. You need to also check that you are comparing the same things here: results from Eulerian TINs should be the same average as Lagrangian TINS provided the Lagrangian values have been re-corrected for advection and strain.

Thank you. The figures and text have been reordered for improved continuity. We include that the Eulerian-derived values are corrected for the effects of advection following Moholdt et al. (2014). As mentioned in the manuscript, we have results for all available Operation IceBridge data in an Eulerian reference frame using a similar TINs-based methodology. You are absolutely correct that there is more available data when computed in an Eulerian frame of reference; however, the data is still spatially sparse over ice shelves (particularly in the pre-IceBridge era). Mission priorities have limited measurements over ice shelves until fairly recently (when Mag/Grav measurements have enabled improved estimations of sub-shelf bathymetry). The strength of the airborne laser altimetry data lies in its accurate measurements at relatively small spatial scales compared to radar altimetry data, repeatable processing methods, and ability to follow glacier flowlines.

3. The authors should look at another Cryosphere Discussions paper by Shean et al. (2018) https://www.the-cryosphere-discuss.net/tc-2018-209/, where Pine Island melt rates are assessed using high-res image-based Lagrangian processing.

We are looking forward to the publication of the Shean et al. paper as it is a very complementary work that uses an independent dataset. As the paper is not presently through peer-review, we have only included citations to the Discussions paper in anticipation of a future acceptance.

4. The authors should probably compare their results for Larsen C with the ATM measurements presented in Adusumilli et al. (2018).

Done. Figure and text have been added.

5. Overall, I think this paper fails to exploit the key features of ATM vs satellite-based products. Satellite altimeters and stereo imagery (Shean et al. (2018), and an earlier Dutrieux et al. (2014) paper), make the process easier, but all satellite altimeters lack spatial resolution and radar altimeters struggle near grounding lines and other steep regions. Think about the new science that is available from a carefully compiled ATM data set where all the biases have been corrected for. If there is no new science, the data set is still valuable as it provides independent estimates to compare with the satellite-derived values. In this case, the most obvious value of the data set is as intended for OIB: a continuation of the ICESat laser altimeter record. Why not look at ICESat data as a third, earlier period in the various plots that compare pre-2011 and post-2011 data?

We expand upon the purpose of this paper and the benefits of having an airborne lidar derived dataset for the validation of satellite datasets and model outputs. We are investigating processes that can be derived with the airborne dataset but are leaving that for a future work. A comparison of Operation IceBridge with ICESat laser altimetry is certainly possible if a compiled dataset was publicly available. We have added a comparison with the published radar altimetry estimates from Adusumilli et al..

**MAJOR: SPECIFIC**

**p.1/I.2:** See general comments. The reader needs to know whether you mean Lagrangian or Eulerian ice thickness change, and if the Lagrangian estimates have been re-corrected back to Eulerian.

Emphasis has been added to clarify that these are Lagrangian estimates.

**p.1/I.8-9:** Comments on Larsen C depend on the quality of the SMB and firn models. This sentence suggests that the ice thickness change really is DH/dt, not dh/dt.

Yes, most of the estimates in this work are Lagrangian-derived ice thickness change. Also, in any reference frame, the conversion from volume to mass will be an important aspect. We added an additional sentence noting how SMB uncertainties will directly impact the basal melt rate estimate.

**p.1/l.9-11:** I don't think \*you\* show that Wilkins depends on "short time-scale and upper-ocean processes": the only evidence I see for this is citations to previous work.

Fair point. Attempting to quantify the effects of individual processes is the subject of some future work.

p.1/I.11-12: Again, this is where you'd be better off reporting dH/dt, even if you're deriving it via recorrected DH/Dt. I was surprised that PIG was "thinning" by 40 m/yr, even close to the grounding line. The more important numbers are in comparisons: you want to show actual Eulerian thinning (dH/dt), BMB, and maybe the ice divergence term.

To clarify, we are reporting Lagrangian thickness change rates and melt rates of the Pine Island Ice Shelf and not Pine Island Glacier. Shean et al. (2018) present very similar numbers for the Pine Island Ice Shelf over similar time periods.

**p.2/I.31:** The Shean et al. (2018) TCD paper is another example of Lagrangian processing.**

The Shean et al. paper is an excellent example of a similar methodology and a very complementary to our own work. Citation has been added.

p.3/I.27-29: I don't understand how you remove non-tidal ocean height change for ice-free ocean points from ice-shelf data. Extrapolate under the ice front? Do you get AVISO sea surface height all the way to the ice fronts at all times of ATM surveys, or does sea ice get in the way? What processes do you think the AVISO products are correcting for, or is this a coarse approximation for regional sea level rise?

Yes, it is simply an extrapolation and is a coarse approximation for regional sea level rise. The extent varies based on sea ice. We chose to use a measured estimate in order to include processes that can deviate strongly from the global ocean averages, such as steric effects and self-attraction and loading effects. The use of this correction follows Paolo et al. (2015) that used the same dataset.

**p.4/I.20 ff: You need to explain all the terms in this equation immediately.**

Great point. We've expanded upon each of the terms.

p.5-6 (Results): This would be clearer if you used sub-headers for each ice shelf that you are considering: Larsen C, Wilkins, Pine Island, and Dotson/Crosson. Also, this is a critical place to use symbols regarding ice thickness change: is it Eulerian dH/dt, Lagrangian DH/Dt, or Lagrangian-derived Eulerian dh/dt?

Great suggestion. The results section has been now partitioned into subsections. We added emphasis to note that all results are in a Lagrangian reference frame and the use of the Eulerian frame was for comparison purposes only.

p.5/I.27-28: Sentence starting "These periods" suggest that RACMO2.3p2 ASE055 is only available for specific periods, which then determine the breakdown of ATM into different epochs. Is this true? Regardless, the reader needs to know the period for which this high-res surface processes model is available.

Great point. We added the range of each climate simulation. "We use 5.5km horizontal resolution outputs from a 1979–2016 climate simulation of the Antarctic Peninsula (XPEN055, van Wessem et al., 2016) and a 1979–2015 climate simulation of West Antarctica (ASE055, Lenaerts et al., 2018)."

**p.5/I.32-33:** Rignot and Jacobs (2002) is not the right cite for "highest impact on glacial flow dynamics". They just assume that and use it to justify looking at melt rates near the grounding line. There are many more recent papers that might be relevant, e.g., Walker et al. (2007), Gagliardini et al. (2010), probably others.

Fair point. We include citations to updated and more relevant work.

p.6/I.3: Rignot (2002) seems like a strange single choice for citation here.

We expand upon this sentence and add reference to more updated work.

**p.6/I.4-13:** The Dotson/Crosson data are incredibly sparse, which I assume is a consequence of using Lagrangian processing given data density on Fig. 1. So (a) is this a place where higher noise in Eulerian would have been better? (b) Maybe you haven't enough data to learn whether conditions are different from the ICESat-era results of Khazendar et al. (2016)? This points again to using the ICESat-era results as a natural comparison for the more recent ATM.

Spatial sparseness at Dotson/Crosson is largely due to the availability of coincident data and not as much of a function of Eulerian versus Lagrangian reference frames. The plot below shows the same time period as the figure in the main text, but uses an Eulerian approach. Khazendar et al. (2016) used the ICESat/OIB data from Sutterley et al. (2014) as an estimate over the grounded ice and estimates from radar depth sounding and ATM over ice shelves. Creating estimates from ICESat would certainly be possible but we believe outside the purview of this paper.

**p.6/I.17:** The statement "Our Eulerian approach" seems to contradict everything you've said about using a Lagrangian approach. This comparison should be much earlier in "Methods", then you could mention "We began by calculating...using three approaches, ..., ..., and ..., applied to Larsen C. Results (Fig. X) demonstrate that..." Just make sure the figure really does compare Eulerian with re-corrected Lagrangian, or advection-and-strain-corrected Eulerian with Lagrangian.

Text and figures have been reordered for improved continuity and for the clarification of points. We also clarify that the Eulerian values are corrected for advection and strain effects.

**p.6/I.26-27:** Dotson/Crosson data are extremely sparse, and it isn't at all clear that Lagrangian methods are the best approach here.

The lack of data at Dotson/Crosson is more due to the lack of coincident flight lines over the ice shelf. This point has been emphasized.

**p.6/l.29-30:** the statement "would likely not be representative" is probably true, especially for Dotson/Crosson, but needs to be justified, e.g., on the basis of data sparseness.

Point added.

p.6/I.30-32: It isn't really obvious why you need a DEM, specifically from photogrammetry, to use ICESat-2 for dH/dt. It helps with the advection terms and Lagrangian TINs, but maybe you need to set up the idea better, along the lines of "The Lagrangian method is strongly dependent on a detailed understanding of surface topographic features being advected by the ice flow..."

Absolutely correct that digital elevation models are not necessary for deriving elevation change with ICESat-2, particularly in an Eulerian reference frame. Lagrangian reference frames are more difficult, particularly if the ice shelf flow is roughly perpendicular to the satellite tracks, such as in the Antwitharctic peninsula. DEMs would improve the tracking of ice parcels, and decrease data "loss" between different satellite tracks. These points have been clarified in the text. Thank you for the comment.

- **p.7/I.13-22:** (a) This section does not flow well, but it does raise two issues that you haven't really explained well up to now:
  - a) Pros and Cons of radar vs laser. The goal, probably, is change in vertically integrated \*mass\* (or ice-equivalent thickness). With laser, you get true surface height really well, but conversion to mass depends on the firn model. If the snow layer is lighter than you thought, you infer too much mass. With radar, it is complicated by penetration (and footprint size), but on the other hand maybe that's good because the inferred reflecting surface is below the lightest snow. However, you still need the model of firn compaction below the reflector.

Good point. We expand upon the strengths and weaknesses of each dataset. We include a more detailed description of the complications for detecting ice shelf surface height from each dataset.

b) The study hasn't really been set up as well as it could have. This gets back to: Is there really new scientific insight here, or is the goal mainly to provide an independent data set that is of specific value in comparing with satellite-derived ice-shelf changes, specifically laser-based? Either way is good for a paper, with the latter being the justification for OIB anyway. A clearer goal, stated early, might help organize the paper so that results are written around that goal. At the moment the paper reads like you're identifying new science, but the

Results section mainly relies on, or repeats, previous studies, just with a new data set. e.g., Adusumilli et al. (2018) reach the same conclusions regarding Larsen C, except they don't spend much time of the advection and-strain terms, but they do use ATM as validation. Wilkins is interesting, but why not compare ATM tracks with ICESat to get a better sense of pre- and post-ICESat behavior?

Fair point. We expand upon the justification of the research and the manuscript in the introduction. We clarify the purpose of the work and justification of using Operation IceBridge data. Creating a comparable dataset using data from the original ICESat mission is worthwhile and certainly possible; however, we believe outside the purview of this manuscript. We include a comparison with the Adusumilli et al. dataset over the Larsen-C ice shelf. We are working towards the overarching goals of ice-ocean interaction and the downstream effects on the grounded ice. We will also have more in depth interpretation of the results for particular ice shelves in future work.

**TECHNICAL: SPECIFIC**

p.1/I.19-20: (a) I think Rignot et al. (2013) just assumes that ice shelves buttress grounded ice, don't show it. You can't cite every paper that makes that claim. (b) Sentence starting "The thinning..." just repeats the idea of buttressing.

The buttressing effect of ice shelves is fairly well documented and we include citations to modeling efforts to quantify the effect.

**p.2/I.1-2: Again, you're repeating the buttressing argument.**

The purpose of this second sentence is to emphasize how changes in ice shelves affect the grounded ice.

**p.2/I.19:** Abbreviation "WFF" isn't used again, so not needed.**

Done. Thank you.

p.2/I.32: Here you cite Rignot et al. (2017) for MEASURES, but on p.4/I.24 it is Mouginot et al. (2017a).

This has been fixed. Thank you. The Mouginot et al. (2017a) data was used to calculate deviations from mean ice flow.

**p.4/I.28:** This reads like the range of validity for hydrostatic is only the narrow band of 1-8 km from the grounding line. You mean that this region is \*not\* hydrostatic, but that the flexural boundary width is in this range.

Correct. The intent was to describe regions downstream of the 1–8 km wide grounding zones. Changed to "The ice thickness estimates are calculated assuming hydrostatic equilibrium, which should be valid for most areas downstream of the 1–8 km wide grounding zones (Brunt et al., 2010, 2011)."

**General style**, especially in Results: You make a habit of starting paragraphs with "Figure X shows...". This sounds like you have a collection of figures to describe, rather than making figures to fit your narrative.

General style has been updated to first describe each ice shelf and introduce results.

**General Style: "{Name} ice shelf" or "{Name} Ice Shelf"?**

Updated. Thank you for the suggestion.

**p.6/I.2: Why refer to Fig. 8\*b\*, specifically?**

Fair point. This has been fixed. Thank you.

**p.6/l.5: I think this means "two periods - 2002-2010 and 2010-2015 - are shown in"**

Precisely. Thank you.

**p.7/l.8: more precisely "maps of time-varying velocity"**

Changed to "time-variable velocity maps"

**p.7/I.9-12:** Would be good to have cites to each of these products.**

Citations added.

**List of Changes**

[revised manuscript text omitted]

- 27. *Added* "If any measurements from the separate flight line lie within 100m of the advected point, the elevation measurement closest in Euclidean distance to the advected point is compared against the original measurement."
- 28. Added "the initial release of"
- 29. *Deleted* "(GSFC)"
- 30. *Added* "Uncertainties in tidal oscillations were estimated using constituent uncertainties from King et al. (2011)."
- 31. Modified "(Legos; Carrère and Lyard, 2003)" to "(Carrère and Lyard, 2003)"
- 32. *Added* "Regional sea levels fluctuate due to changes in ocean dynamics, ocean mass, and ocean heat content (Church et al., 2011; Armitage et al., 2018)."
- 33. *Deleted* "over ice-free ocean points"
- 34. Modified "(Le Traon et al., 1998; CMEMS)" to "(Le Traon et al., 1998)"
- 35. *Added* "The non-tidal sea surface anomalies are added to estimates of mean dynamic topography, which is the mean deviation of the sea surface from the Earth's geoid due to ocean circulation."
- 36. *Added* "The sea surface anomalies are extrapolated from the valid ice-free ocean values to the ice shelf points following Paolo et al. (2016)."
- 37. Deleted "(IMAU)"
- Modified "We use 5.5km horizontal resolution outputs for the Antarctic Peninsula (XPEN055, van Wessem et al., 2016) and West Antarctica (ASE055, Lenaerts et al., 2018)." to "We use 5.5km horizontal resolution outputs from a 1979–2016 climate simulation of the Antarctic Peninsula (XPEN055, van Wessem et al., 2016) and a 1979–2015 climate simulation of West Antarctica (ASE055, Lenaerts et al., 2018)."
- 39. Modified "(van Wessem et al., 2016, Figure 4)" to "(van Wessem et al., 2016)"
- 40. *Added* "The higher spatial resolution topography improves the modeling of wind-driven downstream effects over ice shelves (Datta et al., 2018)."
- 41. Modified "The absolute precision of the RACMO2.3p2 model outputs has been estimated using field data, such as ice cores and surface stake measurements (van de Berg et al., 2006)." to "The absolute precision of the RACMO2.3p2 model outputs has been estimated using Operation IceBridge snow radar observations, satellite observations of surface melt, and and insitu observations, such as ice cores and surface stake measurements, following Kuipers Munneke et al. (2017) and Lenaerts et al. (2018)."
- 42. *Added* "We assume a 15% uncertainty in surface mass balance and firn height change following estimates from Kuipers Munneke et al. (2017)."
- 43. *Added* " $(M_s)$ "
- 44. Added " $(M_b)$ "

- 45. Added " $(M\nabla \cdot V)$ "
- 46. Modified "(Equation 1, Moholdt et al., 2014)" to "(Moholdt et al., 2014)"
- 47. Modified "Modified Equation 1" to " $\frac{dM_s}{dt} + \frac{dM_b}{dt} M\nabla \cdot V = \frac{\rho_w \rho_{ice}}{\rho_w \rho_{ice}} \left(\frac{Dh}{Dt} \frac{\partial h_{oc}}{\partial t} \frac{\partial h_{fc}}{\partial t}\right)$ "
- 48. Deleted "inSAR-derived"
- 49. Modified "Mouginot et al. (2017a)" to "Rignot et al. (2017)"
- 50. Modified "smoothes" to "smooths"
- 51. *Added* "Deviations from mean ice flow were calculated using annually resolved ice velocity maps derived from synthetic aperture radar and optical imagery (Mouginot et al., 2017a)."
- 52. Added "and uncertainties"
- 53. Modified "is" to "are"
- 54. *Modified* "areas 1–8 kilometers downstream of the grounding zone" to "areas downstream of the 1–8 km wide grounding zones"
- 55. Added "3.1 Larsen Ice Shelves"
- 56. *Added* "The ice shelves draining from the Antarctic Peninsula into the Weddell Sea have undergone some significant changes over the past three decades."
- 57. *Added* "The Larsen-A Ice Shelf collapsed in 1995, and the Larsen-B Ice Shelf partially collapsed in 2002 (Rott et al., 2002, 2011)."
- 58. *Added* "The tributary glaciers once flowing into these shelves accelerated with the loss of the ice shelf abutment (Rignot et al., 2008)."
- 59. Added "Remnant"
- 60. Modified "ice shelves" to "Ice Shelves"
- 61. Modified "thickness change" to "melt"
- 62. Modified "ice shelf" to "Ice Shelf"
- 63. Modified "thickness change" to "melt"
- 64. Modified "ice shelf" to "Ice Shelf"
- 65. *Added* "Any uncertainties in reconstructing the regional SMB will significantly impact the resultant basal melt rate estimate."
- 66. *Added* "We compare our airborne laser altimetry estimate of basal melt rates with a long-term record derived from radar altimetry (Adusumilli et al., 2018)."
- 67. *Added* "We find that the radar-derived estimate is comparable with the laser-derived estimate within uncertainties for most points outside of the grounding zone (Figure 7)."
- 68. Added "3.2 Wilkins Ice Shelf"

- 69. *Deleted* "Figure 7 shows the change in ice thickness (a-b) and estimated basal thickness change rates (c-d) of the Wilkins ice shelf for two 3-year periods from 2008–2011 and 2011–2014."
- 70. Modified "ice shelf" to "Ice Shelf"
- 71. Modified "ice shelf" to "Ice Shelf"
- 72. *Deleted* "The delineations were manually digitized as the ice shelf is heavily crevassed in regions near the ice edge and the bay is often filled with ice mélange (Figure 7)."
- 73. *Added* "Figure 8 shows the change in ice thickness (a-b) and estimated basal melt rates (c-d) of the Wilkins Ice Shelf for two 3-year periods from 2008–2011 and 2011–2014."
- 74. Modified "ice shelf" to "Ice Shelf"
- 75. Modified "thickness change" to "melt"
- 76. Added "3.3 Pine Island Ice Shelf"
- 77. *Added* "The Pine Island Ice Shelf abuts one of the most rapidly changing glaciers in Antarctica (Pritchard et al., 2009; Flament and Rémy, 2012)."
- 78. Modified "thickness change" to "melt"
- 79. Modified "ice shelf" to "Ice Shelf"
- 80. *Added* "In this area that was previously grounded, the average estimated basal melt rates from the flight lines were 70±20 m/yr over 2009–2011 and 54±15 m/yr over 2011–2015."
- 81. Modified "(Rignot, 2002)" to "(Rignot, 2002; Shean et al., 2018)"
- Added "However, some of the changes in basal melt rate over the period could be due to large regional interannual-to-decadal variability (Dutrieux et al., 2014; Paolo et al., 2015; Jenkins et al., 2018)."
- 83. Added "3.4 Dotson and Crosson Ice Shelves"
- 84. Deleted "Ice thickness change rates (a-b) and estimated basal thickness change rates (c-d) of the Dotson and Crosson ice shelves for two periods from 2002–2010 and 2010–2015 are shown in Figure 9."
- 85. Modified "ice shelves" to "Ice Shelves"
- 86. Modified "ice shelf" to "Ice Shelf"
- 87. *Added* "Ice thickness change rates (a-b) and estimated basal melt rates (c-d) of the Dotson and Crosson Ice Shelves are shown in Figure 10 for two periods, 2002–2010 and 2010–2015."
- 88. Modified "thinning rates" to "melt rates"
- 89. Modified "thickness" to "elevation"
- 90. Modified "(Moholdt et al., 2014, Figure 10)" to "(Moholdt et al., 2014, Figure 4)"
- 91. Modified "(Moholdt et al., 2014)" to "(Moholdt et al., 2014; Shean et al., 2018)"

- 92. *Deleted* "Our Eulerian approach uses the same Triangulated Irregular Networks (TINs) technique but keeps the point measurement locations static."
- 93. *Deleted* "The Eulerian scheme is similar to the methods of Pritchard et al. (2012) and Rignot et al. (2013) that used ICESat data and required spatial smoothing of the elevation change rates to filter out the effects of advected surface roughness."
- 94. *Modified* "Moholdt et al. (2014) showed a similar improvement when comparing Lagrangian and Eulerian-derived estimates in bottom melt for the Ross and Filchner-Ronne ice shelves." *to* "Moholdt et al. (2014) showed similar improvements in estimating basal melt rates between Eulerian and Lagrangian processing methods for the Ross and Filchner-Ronne Ice Shelves."
- 95. Modified "ICESat data" to "data from the ICESat mission"
- 96. Modified "Lagrangian tracking of airborne data requires 1) a sufficiently wide scanning swath, 2) accurate flow-line flight planning or 3) dense grid measurements." to "Lagrangian tracking of airborne data requires 1) accurate flow-line flight planning, 2) a sufficiently wide scanning swath, or 3) dense grid measurements."
- 97. *Added* "Flight lines along-flow need to be accurately planned to ensure upstream measurements can be paired with future downstream measurements."
- 98. Modified "ice shelf" to "Ice Shelf"
- 99. Modified "ice shelves" to "Ice Shelves"
- 100. Modified "the airborne data are" to "repeated airborne data is"
- 101. Modified "individual" to "singular"
- 102. *Added* "due to the spatial variability of ice thickness change"
- 103. *Modified* "Satellite altimetry measurements from ICESat-2 (Markus et al., 2017) should help rectify the data limitation problem by providing dense point clouds which could be combined with photogrammetric digital elevation models (DEMs) to create ice shelf-wide thickness change maps." *to* "Satellite altimetry measurements from ICESat-2 (Markus et al., 2017) should help rectify the data limitation problem by providing dense and repeated point clouds. ICESat-2 data could be combined with photogrammetric digital elevation models (DEMs) to create high-resolution ice shelf-wide thickness change maps (Berger et al., 2017; Shean et al., 2018)."
- 104. *Deleted* "A more comprehensive update from the ICESat results of Pritchard et al. (2012) and Rignot et al. (2013) will be possible once ICESat-2 data become available."
- 105. *Added* "Combining ICESat-2 with DEMs would help improve the use of the laser altimetry data in a Lagrangian reference frame as ice parcels could be accurately tracked between separate satellite tracks."
- 106. *Modified* "ice shelf" to "Ice Shelf"
- 107. *Modified* "(Hogg and Gudmundsson, 2017, Figure 5)" *to* "(Hogg and Gudmundsson, 2017, Figure 6)"
- 108. Modified "velocity time series" to "time-variable velocity maps"

- 109. Added "(Fahnestock et al., 2016; Gardner et al., 2018; Mouginot et al., 2017a)"
- 110. Added "Improvements in ice thickness and ice velocity estimates will also greatly improve estimates of flux divergence and as a consequence estimates of basal melt rates calculated using mass conservation (Berger et al., 2017; Adusumilli et al., 2018)."
- 111. *Deleted* "Our study provides a validation dataset for floating ice shelves using high-resolution airborne laser altimetry data."
- 112. *Added* "Idealistically, the laser altimeter will detect the snow surface and the radar altimeter will detect the snow-ice interface."
- 113. *Added* "Because laser altimeters ideally detect the snow surface, an estimate of the total column snow/firn height change is needed to calculate the ice shelf freeboard change (Pritchard et al., 2012)."
- 114. *Modified* "Variations in the dielectric properties of the snow due to variable temperatures and snow grain sizes can affect the radar penetration depth (Rémy and Parouty, 2009)." to "For radar altimeters, the radar penetration depth is affected by variations in the dielectric properties of the surface layer due to variations in temperature, snow grain size, snow density and moisture content (Partington et al., 1989; Rémy and Parouty, 2009)."
- 115. *Added* "Due to the variations in penetration depth, estimates of the firn height change below the detected surface are necessary in order to calculate the freeboard change."
- 116. *Added* "Our study provides a validation dataset for floating ice shelves using high-resolution airborne laser altimetry data (Figure 7)."
- 117. *Deleted* ", the two methods most applicable to airborne data,"
- 118. *Deleted* "Figure 4 (previous)"
- 119. Replaced "Figure 4" with "Figure 10 (previous)"
- 120. Replaced "Figure 5" with "Figure 6 (previous)"
- 121. Replaced "Figure 6" with "Figure 5 (previous)"
- 122. *Added* "Figure 7"
- 123. Replaced "Figure 8" with "Figure 7 (previous)"
- 124. Replaced "Figure 9" with "Figure 8 (previous)"
- 125. Replaced "Figure 10" with "Figure 9 (previous)"

[revised manuscript text omitted]

- Sutterley, T. C., Velicogna, I., Rignot, E. J., Mouginot, J., Flament, T., van den Broeke, M. R., van Wessem, J. M., and Reijmer, C. H.: Mass loss of the Amundsen Sea Embayment of West Antarctica from four independent techniques, Geophysical Research Letters, 41, 8421–8428, doi: 10.1002/2014GL061940, URL http://doi.wiley.com/10.1002/2014GL061940, 2014GL061940, 2014.

- Sutterley, T. C., Velicogna, I., Fettweis, X., Rignot, E., Noël, B., and van den Broeke, M.: Evaluation of Reconstructions of Snow/Ice Melt in Greenland by Regional Atmospheric Climate Models Using Laser Altimetry Data, Geophysical Research Letters, 45, 8324–8333, doi: 10.1029/2018GL078645, URL https://doi.org/10.1029/2018GL078645, 2018.
- van de Berg, W. J., van den Broeke, M. R., Reijmer, C. H., and van Meijgaard, E.: Reassessment of the Antarctic surface mass balance using calibrated output of a regional atmospheric climate model, Journal of Geophysical Research: Atmospheres, 111, doi: 10.1029/2005JD006495, URL http:// dx.doi.org/10.1029/2005JD006495, d11104, 2006.
- van Wessem, J. M., Ligtenberg, S. R. M., Reijmer, C. H., van de Berg, W. J., van den Broeke, M. R., Barrand, N. E., Thomas, E. R., Turner, J., Wuite, J., Scambos, T. A., and van Meijgaard, E.: The modelled surface mass balance of the Antarctic Peninsula at 5.5km horizontal resolution, The Cryosphere, 10, 271–285, doi: 10.5194/tc-10-271-2016, URL http://www.the-cryosphere.net/10/271/2016/, 2016.
- Walker, D. P., Brandon, M. A., Jenkins, A., Allen, J. T., Dowdeswell, J. A., and Evans, J.: Oceanic heat transport onto the Amundsen Sea shelf through a submarine glacial trough, Geophysical Research Letters, 34, n/a-n/a, doi: 10.1029/2006GL028154, URL https://doi.org/10.1029/ 2006GL028154, 2007.

**Antarctic Ice Shelf Thickness Change from Multi-Mission Lidar Mapping**

Tyler C. Sutterley1, Thorsten Markus1, Thomas A. Neumann1, Michiel van den Broeke2, J. Melchior van Wessem2, and Stefan R. M. Ligtenberg2

[revised manuscript text omitted]

---

## Author Response (AR2)

Dear Kenny Matsuoka,

We are very appreciative for the reviews of our manuscript "Antarctic Ice Shelf Thickness Change from Multi-Mission Lidar Mapping." In response to the reviewer comments, we have revised the manuscript to clarify some essential points and add a comparative analyses with Pritchard et al. (2012) and Rignot et al. (2013) for the ice shelves in the Amundsen Sea. The modifications did not change the overall conclusions or results.

In the revision, we include:

- 1. A point-by-point response to the reviewer comments. Responses are *italicized and gray*.
- 2. An enumerated list of the modifications made to the manuscript.
- 3. A copy of the manuscript with the changes noted. (Highlighted with red struck-through text to denote subtractions and blue underlined text to denote additions)
- 4. A final copy of the manuscript with those same changes incorporated.

Regards, Tyler C. Sutterley

**First Reviewer Comments:**

In this second version of the manuscript, the Authors have added a substantial discussion to the text, and have also included several new references. The text has been restructured, which does improve the overall flow of the manuscript. There are, however, some pending issues that should be addressed/clarified in the final version.

Thank you for your helpful review of our revised manuscript. We address your comments point-by-point and update the manuscript accordingly.

• There is an issue that has been raised during the first round of review (Reviewer #2 pointed this out very clearly): The distinction between Eulerian vs Lagrangian melt rates. There are two (main) melt quantities when referring to ice shelf melting in the broad sense: (1) The background melt; the melting that gives the ice shelf its configuration (i.e. thicker at the GL and thinning towards the ice front). This means, there is a great deal of melting occurring in the steady state for the ice shelf to acquire its stable geometry. (2) The excess melt; which reflects the ice shelf loss or variability at the base. When equating this melt with the mass change at the surface we obtain the iceshelf net mass loss or gain. Ultimately, we are interested in estimating #2. By tracking a fixed point on the surface of a flowing ice shelf, and comparing this point at two different epochs (this is a Lagrangian observation), we are detecting both melt signals: the thickness gradient (1) and the temporal change (2). So one needs to deconvolve these two from a purely Lagrangian observation. A Eulerian observation, on the other hand, is detecting #2. This is why it is inconsistent to compare directly Lagrangian vs Eulerian observations.

These are good points and points towards the purpose of comparing multiple time periods. By looking at more than span of time, we are able to compare snapshots around the baseline melt. In the future with ICESat-2 and beyond, we can extend the time series and looking at more sub-periods to help isolate both the modern-day background melt, and determine deviations from the background due to climatic variations and possibly long-term change.

• The authors may have addressed this properly, but it is (still) not clear in the text when referring to "Lagrangian melt" what signal is being referred to: the total melt (#1 + #2) [Dh/Dt], which is inherent to the Lagrangian observation; or the Lagrangian-derived melt in excess (#2)  $[dh/dt_{\text{Lagrange}}]$ , which requires posterior separation. For example, in Figure 4, are all the 3 upper panels referring specifically to the temporal change of surface lowering/melt in excess (i.e.  $dh/dt_{\text{Euler}} \text{ vs } dh/dt_{\text{Lagrange}} \dots$  or it is  $dh/dt_{\text{Euler}} \text{ vs. } Dh/Dt$ )?

We update the text accordingly to clarify that we are looking at variations in DH/Dt. We are not determining deflections from the background as we do not have a long enough (30 years) baseline of observations to adequately determine the mean.

• I think this issue still needs to be clarified in the text. It is a complex, and some times nonintuitive, matter. This is a good opportunity for a manuscript addressing both types of measurements to shed some light into it.

This is a great point. We add some context to the discussion section.

• Please increase the dots/symbols on all your plots. Since your estimates are sparse (due to the nature of OIB sampling), it is far more informative to identify the color/value and location of the plotted estimates than trying to stay true to the spatial scale of the measurements (i.e. footprint size of flight lines).

Fair point. The symbol size has been increased.

 "We find that our method is a significant improvement over Eulerian-derived estimates that require substantial smoothing or spatial averaging of the data." (Isn't "smoothing or spatial averaging" the same?) This broad statement is only true because you are dealing with very sparse along flight data. In the presence of good spatial coverage (as that provided by satellites) and high-quality velocity fields, a Eulerian approach might be preferred over a Lagrangian derivation since the latter leads to massive data loss and potentially misses the GL.

Fair point. Intent was to differentiate between smoothing (e.g. Gaussian averaging) and reducing the spatial resolution (i.e. increasing the grid step size). We modify the sentence as the impact of either technique is the same. Yes, idealistically Eulerian-derived thickness change would be preferred. We add more context to the discussion section.

**Minor edits:**

**p1, I3:** Operation IceBridge → NASA's Operation IceBridge**

References have been updated

p1, I3: oceanic and surface processes, using ....

Changed to "oceanic processes from measurements and models, surface velocity measurements from synthetic aperture radar, and high-resolution outputs from regional climate models."

**p5, I3:** "The absolute precision of the RACMO2.3p2 model outputs has been estimated...", and what's the precision?

Kuipers Munneke et al. (2017) list the SMB uncertainty as 15% uncertainty of the SMB rate.

**p9, I5:** "to test their coherence". In the formal statistical sense, "coherence" is correlation as a function of frequency, which I don't think is what you are referring to.

Fair point. Changed to "correspondence".

**Second Reviewer Comments**

**SUMMARY**

The authors use airborne laser altimetry (from airborne topographic mappers (ATM)) over Antarctic Peninsula (AP) and Amundsen Sea (AS) ice shelves, plus models of surface mass balance and firn compaction, to measure ice shelf thinning rates and assign these rates to individual terms in the mass balance. The study is complementary to several previous studies that used satellite altimeters. The coverage of ATM is poor prior to Operation Icebridge (OIB). However, it has some advantages in terms of dedicated tracks, in particular allowing measurements to get close to grounding lines. It is therefore a valuable study, and dataset, to provide to the community.

Thank you Dr. Padman for your helpful second review of our manuscript. We have further revised the manuscript following your suggestions to clarify some essential points and to improve the overall analysis.

**RESPONSES TO ANONYMOUS REFEREE #1**

• "We did not compare with Pritchard et al. (2012) as the data is not provided in a compiled form. Rignot et al. (2013) do not provide publicly available data." This is true; however, data sets are available from these authors on request.

Good point. We updated the manuscript to include comparisons with Pritchard et al. (2012) and Rignot et al. (2013) for the ice shelves in the Amundsen Sea.

 "However, laser altimetry datasets have more accurate surface determination and can more accurately track over regions of abrupt topographical change. ICESat-2 should provide a valuable extension to the laser altimetry record and help separate short term oscillations with longterm change."

It is true that lasers track the true surface better (much better!) than radar. However, if your firn density model (providing the correction for firn air content) is wrong, it is possible that the radar provides a \*better\* estimate of basal mass balance than you get from laser.

This is an excellent point. We expand upon this in the discussion to clarify the strengths and weaknesses of each instrument.

**GENERAL COMMENTS**

The authors have carried out a major overhaul of their manuscript in response to the first round of reviewer comments, including much better organization. However, I still have issues that I think need to be addressed.

1. Figures are not ordered correctly. This made it hard to follow at some points.

We reworked the text to improve continuity for the figures.

2. As I think I now understand, \*all\* thickness change rates are cited in Lagrangian terms. Even "Eulerian TINs" have been corrected for divergence. However, the authors need to appreciate that at least some of their readers are going to default to "thickness change means Eulerian" (i.e., evidence that mass/volume of the ice shelf is changing). I still contend that the "standard" use of Lagrangian methods is to smooth out the individual estimates of height change before \*removing\* the divergence term to get back to thickness change in Eulerian terms, rather than reporting on

Lagrangian changes where all of it \*might\* be divergence with no SMB and BMB contributions. Hence, I still argue for using the Lagrangian derivative symbol (DH/Dt) rather than the words "thickness change", so readers are constantly reminded what they are looking at.

These are good points. We update the passages accordingly.

3. Related to this: One way in which Lagrangian methodology "smooths" thickness change is when changes due to divergence dominant over SMB and BMB variability. That is, even ignoring surface topographic variability that is subsequently advected to create "noise" in the method. Lagrangian processing might produce a smoother field just because the thickness change numbers are larger and more coherent. I'm not arguing against Lagrangian processing, but the manuscript should explain in more detail the effects of different processing options.

While it is true that there may be an inherent smoothing due to the dominance of the flux divergence term, the Lagrangian processing does make a major difference. The plot below is surface elevation from 2008-10-26 compared with a flight line from 2016-11-10 (not corrected for oceanic or surface processes and not corrected for strain). When using Eulerian processing, the oscillations in ice thickness advect "out of phase" and can cause large artificial anomalies when calculating elevation change (difference between Red and Green). With Lagrangian processing, the effect of these gradients is minimized as the same parcels of ice can be compared (difference between Red and Purple).

4. Minor general comment: consistent units (either m or meters, not changing), and space between values and units (200 m, not 200m)

Done.

**SPECIFIC**

**p.1/I.8–9:** See general comments. The first part of this sentence sort of makes sense in terms of Lagrangian DH/Dt, but is then violated by the second part which says that other processes also play a role. Maybe it is "dominated by flux divergence" but certain times and places show other important terms?

Fair point. We modified the sentence accordingly.

"We find that the Larsen-C Ice Shelf is close to steady state over our observation period with spatial variations in ice thickness largely due to the flux divergence of the shelf. Firn and surface processes are responsible for some short-term variability in ice thickness of the Larsen-C Ice Shelf over the time period."

**p.1/I.21:** I still don't think Rignot et al. (2013) is a good citation for evidence of buttressing. There are many others that focus more on the mechanics of this process rather than just asserting it. The same goes for Shepherd et al. (2003) and Fricker and Padman (2012).

We edited these sentences and added more detail about the technical mechanics. "Floating ice shelves can exert control on the grounded ice sheet's overall stability by buttressing the flow of the glaciers upstream (Dupont and Alley, 2005). The response of inland glaciers to ice shelf variations is complicated, and is dependent on both the inland bed topography and the ice shelf geometry (Goldberg et al., 2009; Gagliardini et al., 2010; Gudmundsson, 2013)."

p.1/I.21–24: This would flow better if you started with something like "The mass budget of an ice shelf is the sum of several mass gain and loss terms (Thomas, 1979). Mass is gained by, ... Losses are associated with ... "

Done.

**p.2/I.1:** Isn't mass rather than volume the important term? I think Paolo et al. (2016) used volume because of concerns over firn models, but the other two papers there are attempting \*mass\* balance calcs.

Updated.

p.2/I.4: "at accelerated rates FOR SEVERAL years following the collapse"

Done.

p.2/I.7–9: I think I pointed this out last time: It's the \*increase\* of CDW heat that would drive accelerated thinning, not just the presence of CDW heat. Unless you think it wasn't there at all, the last time these glaciers were in balance, or you are referring to changes \*after\* the irreversible onset of MISI (in which case you need more words.)

Fair points. Jacobs et al. (2011) note that it is likely due to both an increase in CDW heat content and an increase in sub-shelf cavity circulation. We updated the passage accordingly. "The dynamical change of these glaciers likely stems from increases in sub-shelf circulation and heat content of warm Circumpolar Deep Water, which enhanced ocean-driven melt causing thinning of the buttressing peripheral ice shelves (Jacobs et al., 2011)."

**p.2/I.18: What is "Icessn" after "Atm"? If it's important, it'll need to stay with the "ATM" name later on.**

Icessn is the name of the Level-2 ATM product. According to Michael Studinger (GSFC) the meaning of Icessn has been lost over time.

p.2/l.28: delete "be in reference to"; okay just to say "converted to the 2014 solution ... "

Done.

**p.3/I.22:** Scambos et al. (2001) is a very early cite for delineation of ice shelf extent**

NSIDC suggests this citation for the MODIS images of ice shelves dataset. The delineations were done in this study.

**p.3/I.27:** "corrected for ice strain effects following...". Clearer, and more precise, might be to say "have been corrected to Lagrangian thinning rates by adding in the effects of strain." Also, I think Moholdt et al. \*removed\* strain (to get to Lagrangian-processed, \*Eulerian\* dh/dt), rather than adding in strain.

Done.

"Measurements compiled using the Eulerian TINs scheme have been made comparable to the Lagrangian thinning rates by adding the effects of strain using the relation from Moholdt et al. (2014)."

**p.5/I.14–15:** "firn-column heights" is a bit vague. Something like Fig. 3 would have been a good place to outline what everything is. Is this height relative to "pore closeoff depth" (defined as some density?), or the equivalent of "firn air content", or ???

Good point. We update the text to note that we use the firn-column air content.

p.5/I.29: I'm assuming everything is Lagrangian, but saying "the change in ice thickness of" always implies, to me, a change in the volume/mass of the ice shelf. But in fact you conclude that it's fairly close to steady state, and these 'changes' are just because ice diverges.

Yes, of course. Modified to fit intent.

p.5/I.30: Cite to Figure 6 is wrong; or at least, Fig. 6 should be moved to Fig. 4, and this cite should be to Fig. 4 (it is the first figure cite after Fig. 3).

We added an initial paragraph in the results section to improve the continuity.

**p.6/I.3:** Again "is thinning" means something different from what you want people to be thinking in your Lagrangian FoR.

Updated.

p.6/l.10: wrong figure cite. (last one l'll point out.)

Figure 7 was the correct cite for this sentence.

**p.6/I.11:** "Wilkens" $\rightarrow$ "Wilkins"**

Fixed.

**p.6/l.14:** Not clear what other way it can "ablate" other than through basal melting. What other explanation do you have in mind, that you are discounting?

Particularly in the Peninsula, there can be surface melt but how melt is routed is pretty uncertain.

**p.6/l.16:** Many readers will not know whether 6000 km2 is a lot or not. Maybe add ", from xx,xxx to yy,yyy km2"**

Done. We also added the percent change

p.6/I.19–21: This is an interesting case where basal melt rate greatly exceeds Lagrangian thinning. But I struggled to understand how it could be, since Lagrangian thinning includes the basal melt. Since DH/Dt